# A tug of war between filament treadmilling and myosin induced contractility generates actin rings

Qin Ni[1], Kaustubh Wagh[2†], Aashli Pathni[3†], Haoran Ni[4†], Vishavdeep Vashisht[4†], Arpita Upadhyaya[2,4,5*], Garegin A Papoian[4,5,6*]

[1]Department of Chemical and Biomolecular Engineering, University of Maryland, College Park, College Park, United States; [2]Department of Physics, University of Maryland, College Park, College Park, United States; [3]Biological Sciences Graduate Program, University of Maryland, College Park, College Park, United States; [4]Biophysics Graduate Program, University of Maryland, College Park, College Park, United States; [5]Institute for Physical Science and Technology, University of Maryland, College Park, United States; [6]Department of Chemistry and Biochemistry, University of Maryland, College Park, United States

*For correspondence:
arpitau@umd.edu (AU);
gpapoian@umd.edu (GAP)

[†]These authors contributed equally to this work

Competing interest: The authors declare that no competing interests exist.

**Abstract** In most eukaryotic cells, actin filaments assemble into a shell-like actin cortex under the plasma membrane, controlling cellular morphology, mechanics, and signaling. The actin cortex is highly polymorphic, adopting diverse forms such as the ring-like structures found in podosomes, axonal rings, and immune synapses. The biophysical principles that underlie the formation of actin rings and cortices remain unknown. Using a molecular simulation platform called MEDYAN, we discovered that varying the filament treadmilling rate and myosin concentration induces a finite size phase transition in actomyosin network structures. We found that actomyosin networks condense into clusters at low treadmilling rates or high myosin concentrations but form ring-like or cortex-like structures at high treadmilling rates and low myosin concentrations. This mechanism is supported by our corroborating experiments on live T cells, which exhibit ring-like actin networks upon activation by stimulatory antibody. Upon disruption of filament treadmilling or enhancement of myosin activity, the pre-existing actin rings are disrupted into actin clusters or collapse towards the network center respectively. Our analyses suggest that the ring-like actin structure is a preferred state of low mechanical energy, which is, importantly, only reachable at sufficiently high treadmilling rates.

## Editor's evaluation

This important paper uses molecular simulations to explain how actomyosin networks transition from small clusters to the cortex or ring-shaped actin networks. The authors provide compelling evidence that variation in filament turnover rate and myosin concentration triggers a phase transition of these networks. The predictions of this model are consistent with observations made in T cells, where actin ring formation can be induced following their activation by antibodies.

## Introduction

In eukaryotic cells, actin filaments and myosin motors self-organize into a diversity of shapes (*Blanchoin et al., 2014*). A shell-like cortex is ubiquitously found under the cell membrane, which is characterized by a mesh-like geometry and plays an indispensable role in defining cellular shape and mechanochemical responses (*Salbreux et al., 2012*; *Stewart et al., 2011*; *Bovellan et al., 2014*). In

immune cells such as T cells, the actin cortex reorganizes into a peripheral quasi-2D actin ring that sequesters different signaling complexes in separate concentric domains upon stimulation by antigen-presenting cells (*Hui and Upadhyaya, 2017*; *Yi et al., 2012*; *Babich et al., 2012*; *Hammer et al., 2019*; *Murugesan et al., 2016*). Ring-like actin geometries have also been widely found in other sub-cellular structures such as podosomes and axons (*Collin et al., 2008*; *Xu et al., 2013*). How actin filaments and associated motors and proteins assemble into such ubiquitous networks that control the shape of living cells and tissues remains poorly understood due to the complex and non-equilibrium nature of actomyosin networks.

In vitro networks reconstituted from purified proteins have been extensively used to derive the minimal set of determining conditions that govern the assembly, growth, and structural properties of actin networks. However, in vitro networks exhibit strikingly different higher order structures as compared to those in cellular networks. In notable contrast to the ring-like or shell-like networks ubiquitously seen in living cells, in vitro experiments primarily result in actomyosin networks comprised of clusters that originate from global geometric collapse due to myosin motor-driven contractility (*Murrell and Gardel, 2012*; *Chuang et al., 2018*; *Linsmeier et al., 2016*; *Popov et al., 2016*; *Niu et al., 2017*; *Mak et al., 2016*; *Walcott and Sun, 2010*; *Komianos and Papoian, 2018*). The origins of this disparity likely lies in the quantitatively different parameter spaces occupied by in vitro actin networks as compared to their cellular counterparts.

Actin filaments are highly dynamic, undergoing rapid polymerization and depolymerization, and are subject to contractile forces generated by myosin motors (*Levayer and Lecuit, 2012*; *Fritzsche et al., 2016*). Actin polymerization is polarized: monomeric actin (G-actin) binds to the barbed ends of filaments and polymeric actin (F-actin) dissociates from the pointed ends in a process called treadmilling (*Bugyi and Carlier, 2010*; *Pollard, 2007*; *Ni and Papoian, 2019*). We hypothesized that the differences between the predominant actomyosin architectures formed in vitro versus those observed in vivo may arise from the large difference in the corresponding treadmilling rates: in vitro networks reconstituted from purified proteins exhibit treadmilling rates that are often several-fold slower than those observed in vivo due to the lack of regulators that promote actin filament polymerization and disassembly (*Kovar et al., 2006*; *Malik-Garbi et al., 2019*; *McCall et al., 2019*; *Jansen et al., 2015*; *Reymann et al., 2011*). We further postulated that these differences in treadmilling rates render in vitro networks less resistant to myosin-induced collapse. A systematic way to explore how treadmilling rates and myosin contractility combine to shape actomyosin network architecture is essential to probe our hypothesis. This is a difficult experimental task, requiring careful manipulation of molecular machinery and actin polymerization kinetics. Such limitations can be overcome by computer simulations, which provide a powerful way to capture the complex chemistry and mechanics of the active cytoskeleton, and bring significant mechanistic insights.

In order to find a minimal set of conditions that lead to the formation of rings and cortices, we combined computer simulations via the open-access platform MEDYAN (Mechanochemical Dynamics of Active Networks; *Popov et al., 2016*) and experiments on live T cells. We find that the competition between actin filament treadmilling and myosin contractility determines the overall network morphology. Our simulations showed that the speed of actin filament treadmilling drives the network away from global centripetal actomyosin clustering, resulting instead in centrifugal condensation that creates ring-like and cortex-like structures, without tethering filaments to the boundary. On the other hand, increasing myosin motor activity or decreasing filament treadmilling rates lead to a centripetal collapse of actin networks, creating clusters in the network center. Our corroborating experiments on live T cells and simulations mimicking experimental conditions showed that, indeed, hyper-activating myosin II via Calyculin A (CalyA) or inhibiting filament treadmilling via Latrunculin A (LatA) disassembled pre-existing actin rings, causing the network to condense centripetally, resulting in clusters.

Furthermore, our computational analysis indicates that actin filaments located at the network periphery have lower mechanical energy as compared to those that form actomyosin clusters and hence represent the energetically preferred configuration. However, this energetic state is only achievable at sufficiently high treadmilling rates, while at lower treadmilling rates, the system gets trapped in long-lived states where actin filaments instead condense into clusters. In summary, our work shows that a tug of war between filament treadmilling and myosin-induced contraction determines the fate of actomyosin architectures: the energetically favorable ring/cortex states are kinetically accessible only at higher treadmilling rates. Our findings reveal that the assembly and stability

of various cellular actin structures are crucially regulated by the fine-tuning of filament treadmilling, which can be achieved by the activation of accessory proteins, such as formin, profilin, and cofilin, via local biochemical signaling.

## Results

### Dissecting and modeling the T cell actin ring

In order to construct a molecular model of actin rings, we first examined the F-actin distribution in live Jurkat T cells expressing tdTomato-F-tractin (an indirect reporter of F-actin) and MLC-EGFP (myosin light chain). These cells were allowed to spread on an activating glass surfaces coated with anti-CD3 antibody and imaged with time-lapse total internal reflection fluorescence (TIRF) microscopy to visualize the dynamics of actin reorganization (*Figure 1—video 1*). Upon activation by stimulatory antibodies, the actin cytoskeleton in T cells reorganizes into a ring-like structure characterizing the immune synapse (*Figure 1a-c* and *Hui and Upadhyaya, 2017*; *Yi et al., 2012*; *Babich et al., 2012*; *Hammer et al., 2019*; *Murugesan et al., 2016*). The actin ring consists of an outer lamellipodial region and an inner lamellar ring. In the outer ring, Arp2/3 is activated by WASP near the membrane (*Takenawa and Suetsugu, 2007*), generating a branched actin network that largely excludes non-muscle myosin II (NMII) as shown in *Figure 1c*. The inner ring is enriched in actin filaments decorated with NMII which form actomyosin 'arcs' (*Figure 1b–c*). The central region is largely depleted of actin and NMII.

To understand the biophysical determinants of ring formation and stability, we modeled the formation of actin ring systems using MEDYAN, a simulation platform that combines sophisticated, single molecule level treatment of cytoskeletal reactions, polymer mechanics, and mechanochemical feedback. Actin networks were simulated in a thin oblate cylinder with diameters between 3.8 μm and 10 μm, to mimic the lateral dimensions of small mammalian cells (*Figure 1d*). Model details can be found in Simulation Methods. We first modeled an actin network with Arp2/3 mediated branching near the periphery. Simulations show that this preferential activation of branching alone is sufficient to generate a lamellipodia-like actin ring, similar to the outer T cell ring, without any other cytoskeletal components, or filament tethering to the boundary (*Figure 1—figure supplement 1*). We then added the motor protein NMII, crosslinker alpha-actinin, and a filament nucleator formin, which are essential components for actin network remodeling and are ubiquitously found in actin rings and cortices (*Salbreux et al., 2012*; *Blanchoin et al., 2014*). Arp2/3 creates a dense dendritic actin mesh at the cell periphery (*Svitkina and Borisy, 1999*; *Takenawa and Suetsugu, 2007*), and we hypothesize that NMII is sterically expelled from this region as observed in T cells. To mimic in vivo conditions, we excluded NMII from the peripheral region which contains Arp2/3 mediated branched actin networks. Upon tuning the concentrations of cytoskeletal components and filament treadmilling rates, we found that the network self-organizes into and maintains an outer lammellipodia-like ring and an inner lamellar-like ring with similar actomyosin spatial distribution as the actin ring in T cells (*Figure 1e*). Also similar to T cells (*Hui and Upadhyaya, 2017*; *Babich et al., 2012*; *Yi et al., 2012*), simulated F-actin undergoes retrograde flow due to filament polymerization against the boundary and NMII generated contraction (*Figure 1—figure supplement 2*). Even without spatial restrictions on actin or myosin at the periphery, our simulated networks resemble the inner actomyosin ring found in T cells, suggesting that the formation of a ring-like actin structure is a consequence of actomyosin self-organization. We next focused on the origins of this inner actomyosin ring.

### Building a minimal model for actin ring formation

To explore the minimal determinants of actin ring formation, we first modeled networks with only actin filaments at different average treadmilling rates ($\langle r_{TM} \rangle$ = 0.57 s$^{-1}$, 1.41 s$^{-1}$, and 2.21 s$^{-1}$) based on actin filament assembly kinetics reported from prior experiments (*Fujiwara et al., 2007*; *Kovar et al., 2006*; *Fujiwara et al., 2018*). These systems also include formin at a concentration of 100 nM (*Pring et al., 2003*; *Ni and Papoian, 2019*). We found that disordered actin networks were created at all treadmilling conditions tested (*Figure 2—figure supplement 1*, b). We quantified the spatio-temporal evolution of the network geometry by plotting the median of the radial filament density distribution ($R_{median}$) as a function of time (*Figure 2—figure supplement 1*, a). In NMII-free networks, we observed a relatively uniform filament density across the network regardless of treadmilling rates

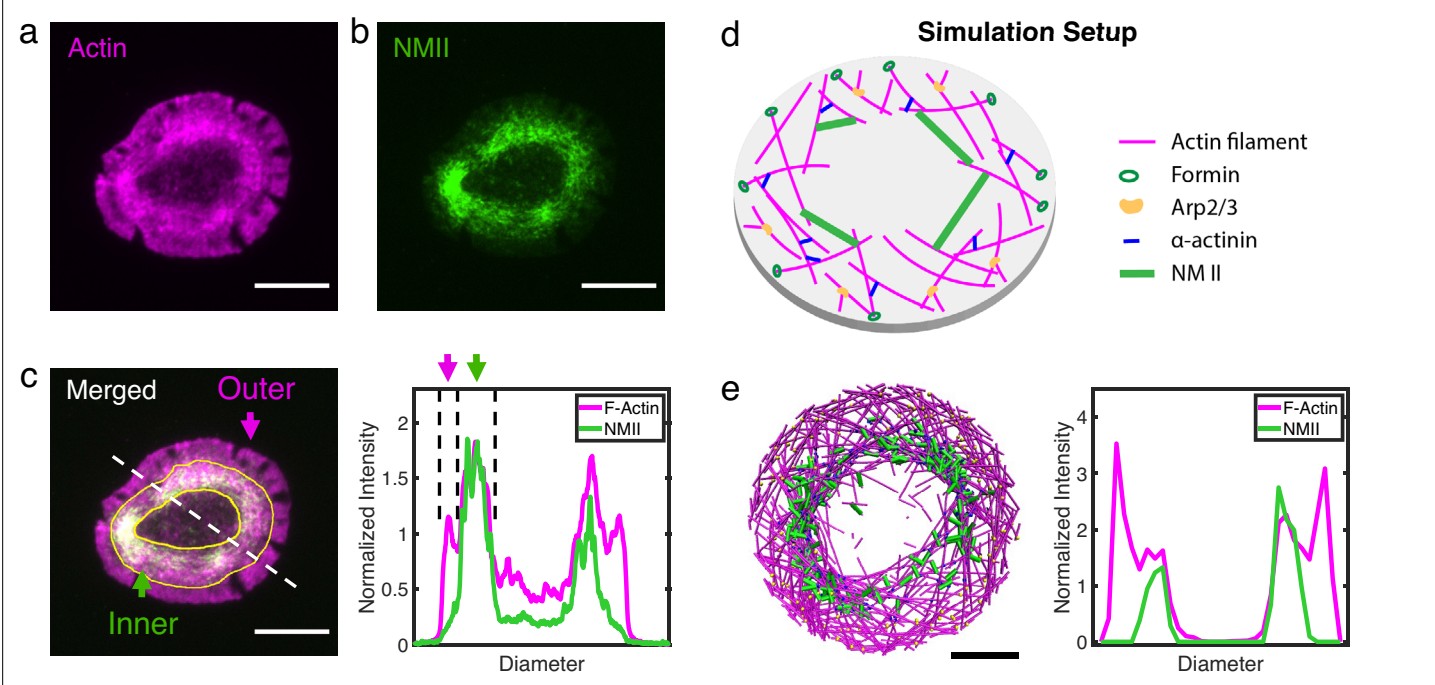

**Figure 1.** Actin and NMII distribution within actin rings in T cells and simulation. (**a–b**) Representative snapshots of actin (**a**) and NMII (**b**) in actin rings of live Jurkat T cells activated on anti-CD3 antibody-coated coverslips. Actin is labeled by tdTomato F-tractin (magenta), and NMII is labeled by MLC-EGFP (green). (**c**) Merged fluorescence image (left panel) showing distribution of actin and NMII within the T cell actin ring. Inner and outer regions of the ring are indicated. Normalized fluorescence intensity profiles of F-actin and NMII (right) along the dashed line shown in the left panel. (**a–c**) Scale bar = 10 μm. (**d**) Setup of simulations using MEDYAN with the major cytoskeletal components labeled. (**e**) A representative snapshot of the simulation (left)and the corresponding distribution of actin and NMII along the diameter of the ring (right). $C_{actin} = 120\mu M, C_{NMII} = 0.1\mu M, C_{alpha-actinin} = 4\mu M, C_{Arp2/3} = 1\mu M, C_{formin} = 0.3\mu M$. Scale bar = 1μm.

The online version of this article includes the following video and figure supplement(s) for figure 1:

**Figure 1—video 1.** Timelapse movie of F-actin and NMII in Jurkat T cells activated by anti-CD3 coated stimulatory coverslips as shown in Figure 1a. https://elifesciences.org/articles/82658/figures#fig1video1

**Figure supplement 1.** Representative snapshots of actin networks that consist of 40μM actin and 1–4μM Arp2/3 without crosslinkers or motors.

**Figure supplement 2.** F-actin flow rate along the radial direction.

(*Figure 2—figure supplement 1*, c). In this case, the network geometry is dominated by stochastic filament treadmilling that is not spatially biased. The boundary plays an important role, as the boundary repulsion force inhibits barbed end polymerization such that filaments reaching the boundary rapidly depolymerize and eventually disassemble. The loss of filaments through depolymerization is compensated by the nucleation of new filaments, resulting in dynamic and disordered structures (*Figure 2—video 1*).

We next explored how these disordered networks behaved upon the introduction of crosslinking and motor contractility. We allowed the network to evolve for 300 s at different $\langle r_{TM} \rangle$ as described above to reach a steady disordered state, and then added NMII and the actin crosslinker alpha-actinin to generate contractile forces. The addition of NMII and crosslinkers changed the steady state network geometry, as measured by $R_{median}$ (*Figure 2a*). For slow treadmilling rates ($\langle r_{TM} \rangle = 0.56s^{-1}$), the addition of NMII and alpha-actinin resulted in the clustering of actin filaments (*Figure 2b–iii*, and *Figure 2—video 1*). This geometric pattern is consistent with prior in vitro and in silico studies on contractile actomyosin networks (*Murrell and Gardel, 2012*; *Chuang et al., 2018*; *Linsmeier et al., 2016*; *Niu et al., 2017*), where contractility can be defined as a symmetry breaking event accompanied by a geometric collapse of the network. The average local concentration of actin within the clusters was 234 μM (*Figure 2—figure supplement 2*), which is almost six-fold higher than the initial G-actin concentration (40 μM), suggesting a high degree of condensation. Although the size and

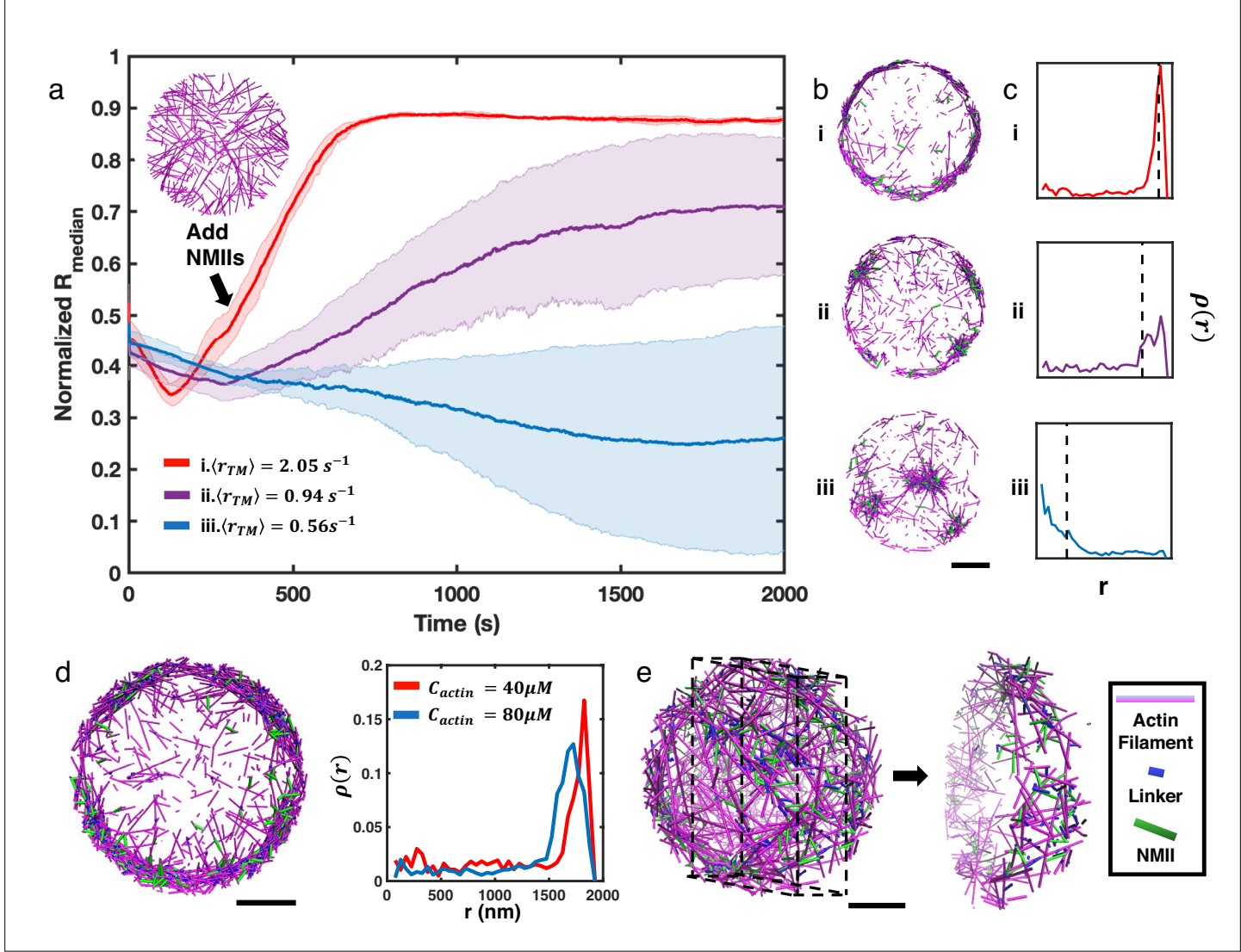

**Figure 2.** NMII contractility induces geometric collapse of treadmilling actin filaments. (**a**) Normalized medians of radial filament density distribution ($R_{median}$) at different treadmilling rates ($\langle r_{TM}\rangle$) are shown. The treadmilling rate is defined as the average number of actin monomers added per filament per second at the barbed ends - equivalent to the rate of F-actin depletion from the pointed ends - after reaching the kinetic steady state (See Simulation Methods and *Figure 2—figure supplement 7* for details). 0.06μM of NMII and 4μM of alpha-actinin were added at 301s. The inset figure is a snapshot at t=300s of networks with $\langle r_{TM}\rangle = 2.05 s^{-1}$. The shaded error bars represent the standard deviation across 5 runs. (**b–c**) Representative snapshots at each treadmilling condition (**b**) and their radial filament density distribution, $\rho(r)$ (**c**) are shown. Dashed lines in (**c**) indicate the position of $R_{median}$. (**d**) Representative snapshot of ring-like networks with 80μM actin (left), and $\rho(r)$ of actin rings with 40μM actin and 80μM actin (right) are shown ($\langle r_{TM}\rangle = 1.35 s^{-1}$). (**e**) A snapshot of a spherical cortex-like network (left) and a slice showing the internal structure (right). (**a,b,d,e**) Actin filaments are magenta cylinders, NMIIs are green cylinders and linkers are blue cylinders in all snapshots. All scale bars are 1μm.

The online version of this article includes the following video and figure supplement(s) for figure 2:

**Figure 2—video 1.** The simulated actin network either contains only actin filaments (top left), as shown in Figure 2—figure supplement 1, or contains actin filaments, myosin, and crosslinkers (top right, bottom left and right) with average tradmilling rate  and , respectively, as shown in Figure 2a–b.
https://elifesciences.org/articles/82658/figures#fig2video1

**Figure 2—video 2.** Myosin motors bend actin filaments in fast treadmilling networks.
https://elifesciences.org/articles/82658/figures#fig2video2

**Figure 2—video 3.** The evolution of a fast-treadmilling network in a spherical boundary as shown in Figure 2e.
https://elifesciences.org/articles/82658/figures#fig2video3

**Figure supplement 1.** Actin networks remain disordered in the absence of myosin, regardless of treadmilling rate.

**Figure supplement 2.** Heatmap showing a pixellated representation of the local F-actin concentration in 100nm x 100nm bins.

*Figure 2 continued on next page*

*Figure 2 continued*

**Figure supplement 3.** Five trajectories of medians of filament radial density distribution and simulation snapshots with $\langle r_{TM} \rangle = 0.56 s^{-1}$.

**Figure supplement 4.** The distribution of filament orientations for disordered networks and ring-like networks near the network periphery .

**Figure supplement 5.** Comparison of network evolution under different treadmilling conditions in a larger, 10 μm geometry.

**Figure supplement 6.** Evolution of three dimensional actin networks under different treadmilling conditions.

**Figure supplement 7.** Actin filament length distributions under different treadmilling conditions.

location of actin clusters varied significantly across multiple trajectories (*Figure 2—figure supplement 3*), a decreasing $R_{median}$ suggests that the overall collapse is centripetal (*Figure 2b–iii*).

Our simulations suggest that actin networks are subject to two competing processes: treadmilling, which tends to homogeneously distribute filaments in the network, and NMII-mediated contractility, which tends to trap filaments into clusters. We thus explored changes in the actin network geometry by increasing the treadmilling rate while maintaining the same concentration of NMII. Although filament nucleation occurs stochastically throughout the entire network and there is no filament tethering near the boundary, we discovered that after addition of NMII to rapidly treadmilling networks ($\langle r_{TM} \rangle = 2.05 s^{-1}$), filaments steadily accumulate at the network boundary (*Figure 2a–i*, and *Figure 2—video 1*). During this process, we observed that NMII deformed many filaments and gradually changed their orientation from being perpendicular to the boundary to parallel (*Figure 2—video 2*). Upon allowing the system to further evolve for several hundred seconds, we found that actin networks transformed into ring-like structures (*Figure 2b–i*). Networks with intermediate $\langle r_{TM} \rangle = 0.94 s^{-1}$ form a mixture of clusters and rings (*Figure 2b–ii*, and *Figure 2—video 1*).

The resulting actin rings are highly condensed, with a thickness of a few hundred nanometers and exhibiting local actin concentrations similar to those found in actin clusters (263 μM). Increasing the initial G-actin concentration increases the thickness of actin rings (*Figure 2d*). Most filaments in actin rings are oriented parallel to the boundary (*Figure 2—figure supplement 4*), forming small actin clusters that undergo azimuthal flow (*Figure 2—videos 1; 2*). Analogous ring-like patterns were observed on a larger system with a diameter of 10 μm (*Figure 2—figure supplement 5*). In a spherical system, networks evolved into hollow spherical cortex-like geometries under similar conditions (*Figure 2e*, *Figure 2—figure supplement 6*, and *Figure 2—video 3*).

## Competition between filament treadmilling and NMII contractility determines network morphology

To further examine how treadmilling rate regulates the formation of distinct actomyosin architectures, we performed extensive simulations at different treadmilling rates. Indeed, $\langle r_{TM} \rangle$ emerges as a key control parameter that governs the steady state network geometry. Below a critical $\langle r_{TM} \rangle$, which is 0.94 $s^{-1}$ in our simulations, networks geometrically collapse into clusters, while above this critical $\langle r_{TM} \rangle$, they preferentially evolve into ring-like geometries (*Figure 3a and b*). The radial distribution of the ring state is characterized by higher $R_{median}$ and smaller standard deviation compared with the cluster phase. Interestingly, $R_{median}$ as a function of $\langle r_{TM} \rangle$ displays a sharp increase as the network transitions from the cluster state to the ring state (*Figure 3b*). Because the $R_{median}$ trajectories after adding NMIIs are almost linear before reaching a steady state, we quantified the network remodeling speed by measuring the slopes of the linear part of the $R_{median}$ trajectories. We found that the network remodeling speed is positively correlated with $\langle r_{TM} \rangle$ (*Figure 3c*), indicating that $\langle r_{TM} \rangle$ is an important factor driving network structural evolution.

We next varied the NMII concentration ($C_{NMII}$) at different treadmilling rates, obtaining a phase diagram delineating actin network morphologies (*Figure 3d–e*). The phase diagram indicates that the critical $\langle r_{TM} \rangle$ for actin ring formation increases as $C_{NMII}$ increases. Networks collapse into clusters for $\langle r_{TM} \rangle$ below the critical value. The higher $C_{NMII}$ is, the more likely a cluster tends to localize to the geometric center of the network, indicating that NMII induced contractility drives the centripetal condensation. When $\langle r_{TM} \rangle$ and $C_{NMII}$ are both low, the network becomes disordered (for example, see *Figure 3e*, $\langle r_{TM} \rangle = 0.56 s^{-1}$ and $C_{NMII} = 0.02 \mu M$). Similarly, increasing network contractility by tuning alpha-actinin concentration also results in a transition from ring-like networks at low linker concentrations to bundles and clusters at higher concentrations (*Figure 3—figure supplement 1*).

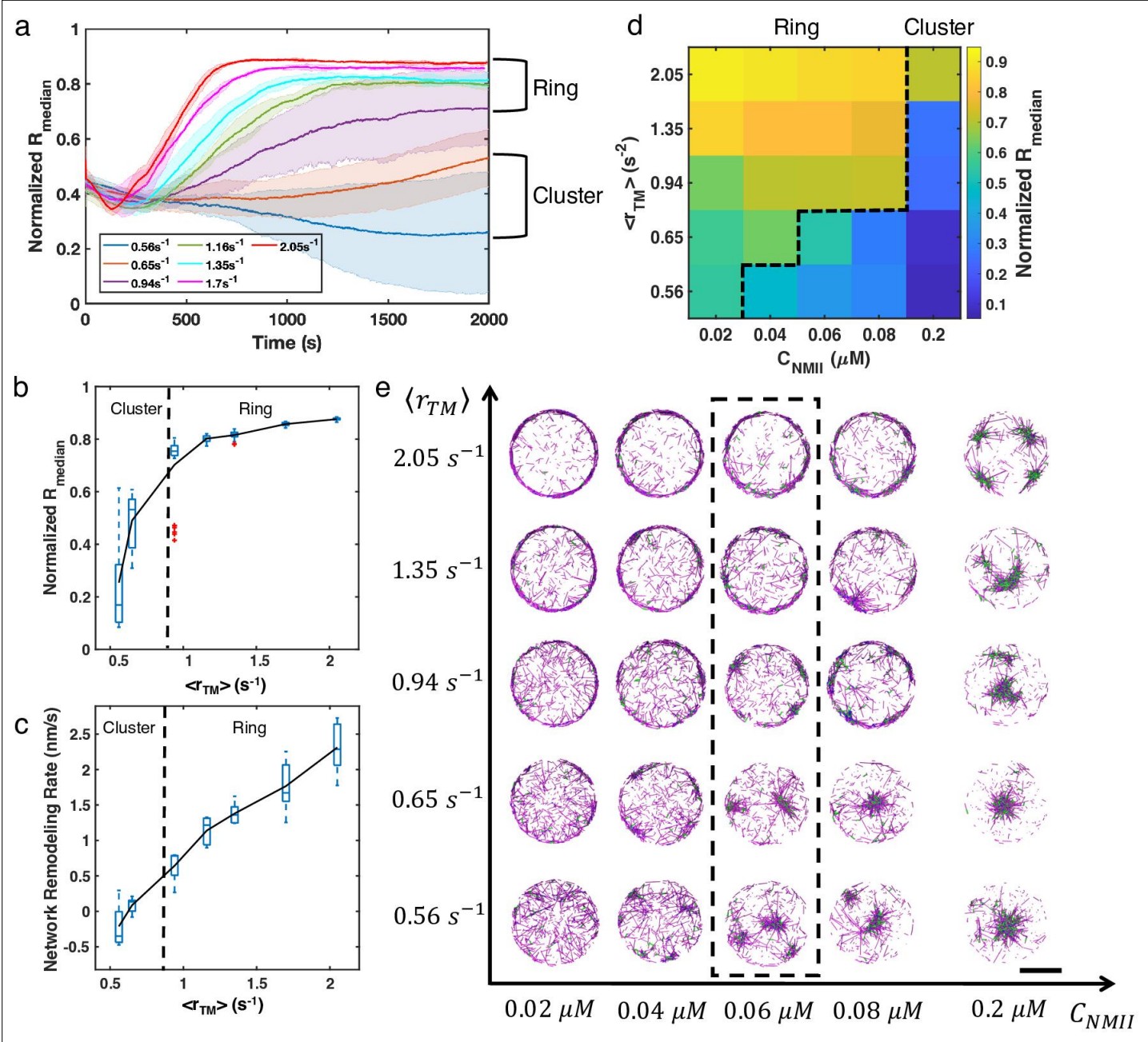

**Figure 3.** Treadmilling rate and NMII concentration regulate network structure transitions. (**a**) Normalized medians of radial filament density distribution ($R_{median}$) as a function of time at different $\langle r_{TM} \rangle$(0.56 $s^{-1}$ to 2.05 $s^{-1}$) are shown. The shaded colors represent the standard deviation of means for 5 runs. (**b**) The box plot shows the average $R_{median}$ in the last 500 s of simulation at each treadmilling rate. Solid line connects the mean $R_{median}$ at each $\langle r_{TM} \rangle$. (**c**) The box plot shows the speed of network remodeling, measured as the slope of the linear part of $R_{median}$ after 300 s. The solid line connects the mean remodeling rates at each $\langle r_{TM} \rangle$. (**a–c**) $C_{actin} = 40\mu M$, $C_{NMII} = 0.06\mu M$, $C_{alpha-actinin} = 4\mu M$. (**d**) Steady state $R_{median}$ at different $\langle r_{TM} \rangle$(0.56 $s^{-1}$ to 2.05 $s^{-1}$) and $C_{NMII}$ (0.02–0.2 μM). (**e**) Representative snapshots of steady state actin network structures at different $\langle r_{TM} \rangle$ and $C_{NMII}$. Representative snapshots of trajectories in (**a**) are shown in the dashed box. (**d-e**) $C_{actin} = 40\mu M$, $C_{alpha-actinin} = 4\mu M$. Actin is depcited as magenta cylinders and NMII as green cylinders. Scale bar = 2 µm.

The online version of this article includes the following figure supplement(s) for figure 3:

**Figure supplement 1.** Representative snapshots of steady state actin network structures at different $C_{alpha-actinin}$ and $C_{NMII}$ for all conditions.

## Inhibition of actin dynamics disrupts actin rings in live cells and in silico

In order to further understand how treadmilling regulates ring-like actin networks, we experimentally disrupted F-actin dynamics in live Jurkat T cells expressing EGFP-F-tractin. Since it is not feasible to directly control the treadmilling rate in experiments, we used the actin inhibitor, Latrunculin-A (LatA), which decreases the polymerization rate and increases the depolymerization rate by sequestering G-actin and accelerating phosphate release from ADP-Pi-actin (*Lodish, 2000*; *Yarmola et al., 2000*; *Fujiwara et al., 2018*). Upon the formation of the actin ring at the contact zone, LatA (at different concentrations) was added to spreading cells and the resulting effect on the rings was monitored

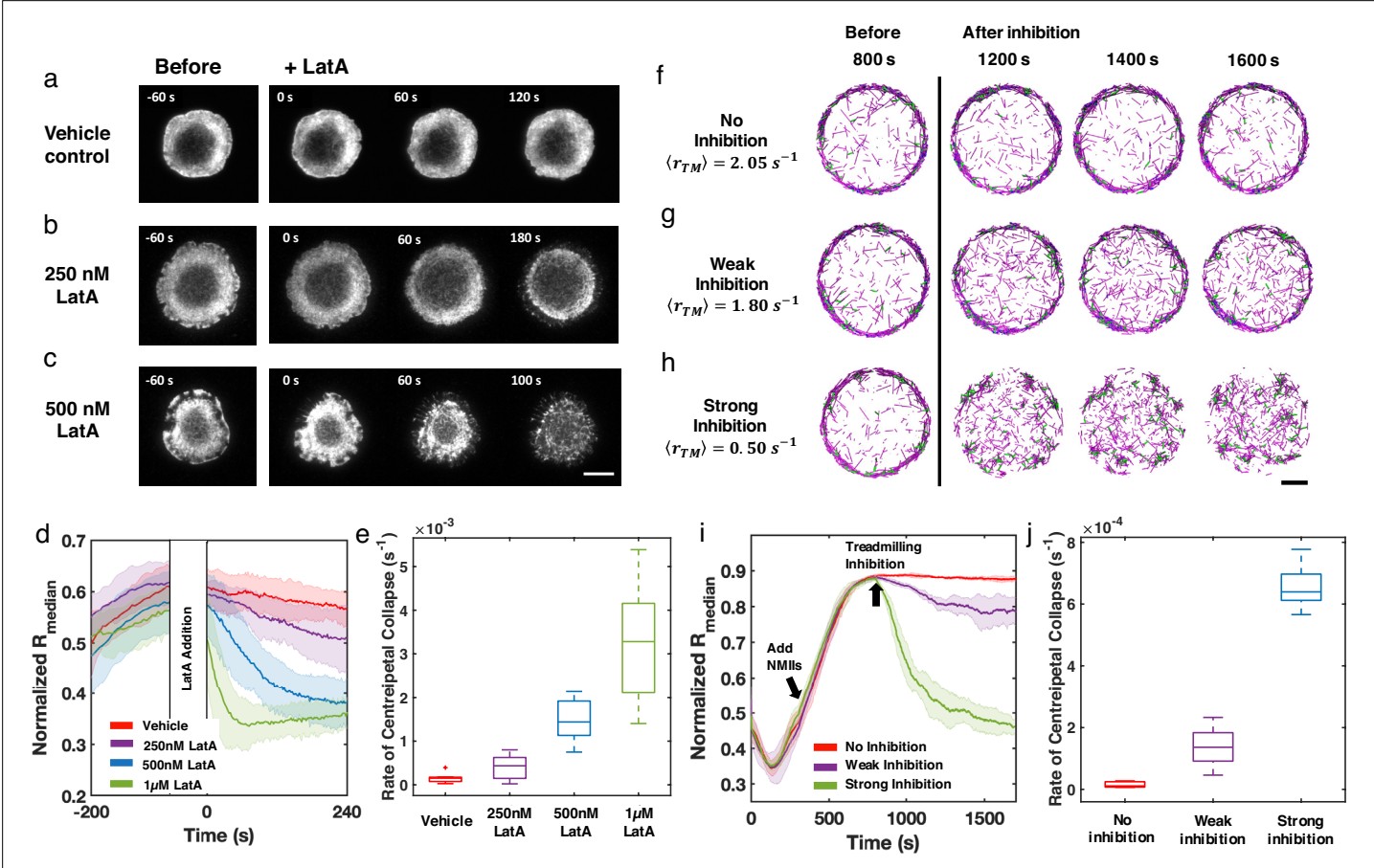

**Figure 4.** Inhibition of actin dynamics induces collapse of actin rings in live T cells and in silico. (a–c) Timelapse montages of Jurkat T cells expressing F-tractin-EGFP spreading on anti-CD3 coated glass substrates. Cells were treated with (a) vehicle control (0.1% DMSO), (b) 250 nM LatA, or (c) 500 nM LatA between 300 and 360 s after contact with activating surface. The first post-treatment image is labeled as 0 s. Timelapse images illustrate the centripetal collapse of the actin ring upon treatment with LatA. Timescales of this collapse depend on the concentration of LatA as can be seen from the timestamps on the images. Scale bar is 10 μm. (d) Quantification of the spatial organization of the actin network using the normalized median of radial filament density distribution. Shaded error bars represent the standard deviations across trajectories (7–11 cells per condition). (e) Box plots showing the rate of centripetal collapse, measured as the slope of the $R_{median}$ distribution after inhibition. (f–h) Timelapse montages of simulations of (f) control, (g) weak inhibition, and (h) strong inhibition. Treadmilling rates in these conditions are $2.05 s^{-1}$, $1.80 s^{-1}$, and $0.50 s^{-1}$, respectively. Indicated $\langle r_{TM} \rangle$ is the averaged treadmilling from 1500 s to the end of simulations. Scale bar indicates 1 μm. (i) Medians of radial filament density distribution at different conditions. (j) Rate of centripetal collapse, measured as the slope of the $R_{median}$ distribution after inhibition. (i, j) The shaded color and error bars represent the standard deviation across trajectories, n=5 runs per condition.

The online version of this article includes the following video for figure 4:

**Figure 4—video 1.** Timelapse movie of F-tractin-EGFP labeled F-actin in Jurkat T cells activated by anti-CD3 coated stimulatory coverslips with 0.1% DMSO (vehicle control), and with 250 nM, 500 nM, and 1 μM LatA, respectively.

https://elifesciences.org/articles/82658/figures#fig4video1

**Figure 4—video 2.** Simulation of actin assembly disruption in ring-like actin networks to mimic LatA inhibition in experiments.

https://elifesciences.org/articles/82658/figures#fig4video2

with time-lapse imaging. In order to compare with simulations, we used the fluorescence intensity as a reporter of F-actin levels and calculated a normalized $R_{median}$ to quantify the evolution of the actin network under varying degrees of LatA inhibition compared to vehicle control (*Figure 4a*). With weak inhibition ($C_{LatA}$ = 250 nM), the ring like structure is perturbed but largely preserved for several minutes (*Figure 4b*, and *Figure 4—video 1*). At higher doses of LatA ($C_{LatA}$ = 500 nM and 1 μM, *Figure 4c*, and *Figure 4—video 1*), $R_{median}$ rapidly decreases (*Figure 4d*), indicating a collapse of the network towards the geometric center of the cell. The rate of centripetal collapse of the actin network increases with increasing $C_{LatA}$ (*Figure 4e*). The dismantling of the actin ring is also accompanied by the formation of F-actin clusters or bundles (*Figure 4b–c*).

To compare with these experimental observations, we perturbed actin network assembly in silico after ring-like networks were established. Based on recent work on reconstituted actin networks under LatA treatment (*Fujiwara et al., 2018*), we reduced the polymerization rate constants and increased the depolymerization rate constants to mimic the effect of LatA on sequestering G-actin and accelerating depolymerization to closely model the T cell experiments. Actin rings (no inhibition, *Figure 4f*) were created in the same way as shown in *Figure 2a–i*. Upon the formation of stable actin rings at 800 seconds, we perturbed actin filament polymerization to different extents to mimic weak and strong LatA inhibition (*Figure 4—video 2*). We found that actin rings persist under weak inhibition (*Figure 4g*), while they collapse into clusters under strong inhibition (*Figure 4h*). The disruption of actin filament assembly also reduces $\langle r_{TM} \rangle$ from $2.05 s^{-1}$ to $1.80 s^{-1}$ (weak inhibition) and $0.50 s^{-1}$ (strong inhibition), respectively. Measurements of $R_{median}$ and the rate of collapse (*Figure 4i and j*) at different inhibition conditions reveal the centripetal collapse of the ring network, reproducing the above-described experimental observations.

## Enhancement of NMII activity leads to centripetal contraction of actomyosin rings in T cells and *in silico*

In order to validate the role of NMII activity in regulating ring-like actin networks, we next altered NMII dynamics in live Jurkat T cells. Under vehicle control (DMSO), actin rings are relatively stable over the timescale of 10 min, and the F-actin distribution displays a steep transition from a depletion zone at the cell center to a high-intensity plateau (*Figure 5—figure supplements 1 and 2a*). Calyculin A (CalyA) application to enhance NMII activity (*Ishihara et al., 1989*) leads to an increase in contractility and a centripetal collapse of the actin network (*Figure 5a*, and *Figure 5—video 1*), as quantified by the decrease of $R_{median}$ (*Figure 5c*). On the other hand, upon treatment with Y-27632, an inhibitor of NMII's upstream regulator, Rho kinase (*Uehata et al., 1997*), which decreases myosin based contractility, the network becomes more disordered and displays a shallower transition from the central depletion zone to the peripheral plateau (*Figure 5b*, and *Figure 5—video 1*). We quantified these changes by calculating the slope of the normalized F-actin intensity from the center to plateau region. As shown in *Figure 5d*, the slope remained constant over time under vehicle addition, while it decreased upon Y-27632 addition, indicating that the network becomes more diffuse and disordered, and the ring integrity is compromised with loss of myosin contractility. These results confirm that NMII is a central regulator of actin network structure, and high NMII activity is antagonistic to actin ring formation.

We then validated the role of NMII in shaping actin structure using MEDYAN simulations. To reduce the computational time, we first initialized actin ring networks and then increased or decreased NMII concentrations (see Materials and methods for simulation setups). Under control conditions, we tuned the actin and NMII concentrations to mimic the conditions tested in T cells (*Figure 5—figure supplement 2b*). In agreement with experiments, enhancing NMII levels induces centripetal collapse of the network (*Figure 5e*, and *Figure 5—video 2*), and the speed of the collapse is proportional to the amount of NMII added to the system (*Figure 5f*). These results also indicate that a confined boundary is not required for the maintenance of actin rings. On the other hand, upon reduction of NMII levels, the actin ring becomes more disordered (*Figure 5—video 2*) and the slope of the center to plateau F-actin distribution decreases (*Figure 5g and h*), in agreement with Y-27632 inhibition experiments.

## Energetic origins of structural polymorphism in active networks

We next explored the chemical and mechanical properties of actin networks at various treadmilling rates. We found that the numbers of F-actin filaments, bound linkers, and bound motors remain nearly

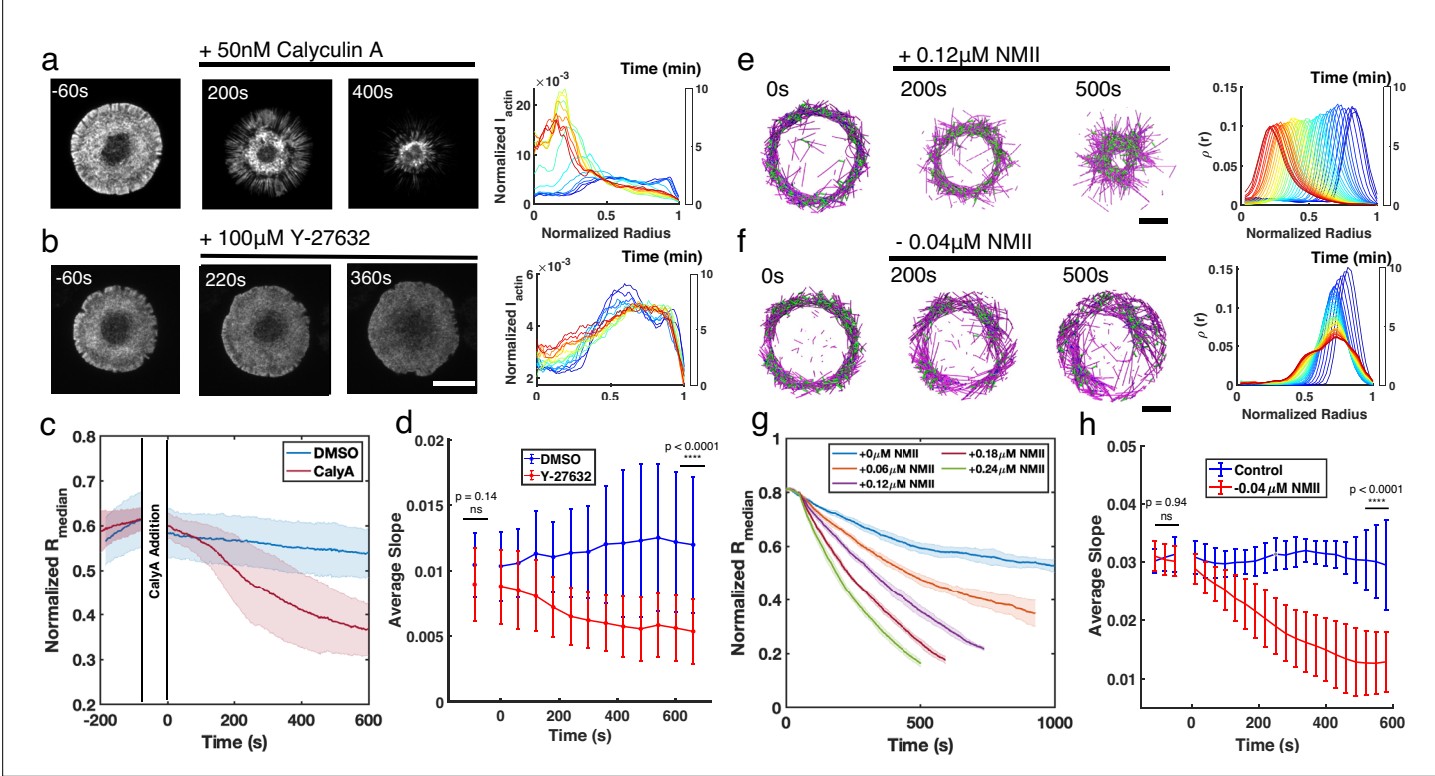

**Figure 5.** Enhancement or inhibition of NMII regulates actin structure in live T cells and in silico. (**a–b**) Time lapse montages of Jurkat T cells expressing F-tractin-EFGP spreading on anti-CD3-coated glass substrates (left) and the normalized radial F-actin intensity (right). After achieving maximal spreading, cells were treated with (**a**) 50 nM CalyA, or (**b**) 100 μM Y-27632. Scale bar is 10 μm. (**c**) The normalized median of radial filament density distribution $R_{median}$. n=12 cells for vehicle (0.5% DMSO), and n=14 cells for CalyA. Two sample t-test was performed for the first point before drug addition (ns, p=0.83) and 600 s after drug addition (****, p<0.0001). (**d**) The slope of the intensity profiles over the transition region from the center to the peripheral plateau as a function of time. n=25 cells for vehicle (0.1% DMSO), and 24 for Y-27832. Two sample t-test was performed for the first point (before drug addition) and the last point (660 s after drug addition). (**e–f**) Timelapse montages of simulations (left) and the normalized radiaul filament density distribution $\rho(r)$ at different times (right) mimicking actin rings in (**e**) CalyA treatment by increasing NMII levels, and (**f**) Y-27632 treatment by reducing NMII levels. An actin ring containing 80 μM actin, 0.18 μM NMII, and 4 μM alpha-actinin was pre-initialized as described in the Simulation Methods, and the NMII perturbation was performed at 0 s. The control condition is shown in *Figure 5—figure supplement 2*. Scale bar is 1 μm. (**g**) The evolution of $R_{median}$ for different levels of NMII addition. Blue curve is control, and other curves are simulations with indicated levels of NMII added to mimic the CalyA experiment. (**h**) The slope of the intensity profiles over the transition region from the center to the peripheral plateau as a function of time for simulations of Y-27632 addition. Blue curve is control while orange curve represents simulations after reduction of NMII concentration by 0.04 μM. Two sample t-test was performed for the first three points (before inhibition) and the last three points (510 s to 600 s after drug addition). (**g–h**) n=5 runs per condition. (**a–h**) In all figures, 0 s represented the first time point recorded after drug addition (for experiments) or NMII addition/depletion (for simulations). (**c,d,g,h**) Shaded colors and error bars represent the standard deviation across cells or simulation trajectories.

The online version of this article includes the following video and figure supplement(s) for figure 5:

**Figure supplement 1.** An example for calculating the slope of center to plateau F-actin distribution.

**Figure supplement 2.** Actin ring dynamics in live T cells and in silico under control conditions in the absence of inhibitions.

**Figure 5—video 1.** Timelapse movie of F-tractin-EGFP labeled F-actin in Jurkat T cells activated by anti-CD3 coated stimulatory coverslips and treated with 50 nM CalyA and 100 μM Y-27632, respectively.

https://elifesciences.org/articles/82658/figures#fig5video1

**Figure 5—video 2.** Simulations of hyper-activating and inhibiting NMII in ring-like actin networks to mimic CalyA and Y-27632 treatment, as shown in Figure 5.

https://elifesciences.org/articles/82658/figures#fig5video2

constant across different $\langle r_{TM} \rangle$, while distributions of diffusive molecules, such as G-actin and nucleators, also did not show spatial localization, being uniformly distributed throughout the simulation volume (*Figure 6—figure supplement 1*). These observations suggest that ring-like architectures do not form because of the enrichment of soluble constituent molecules near the periphery.

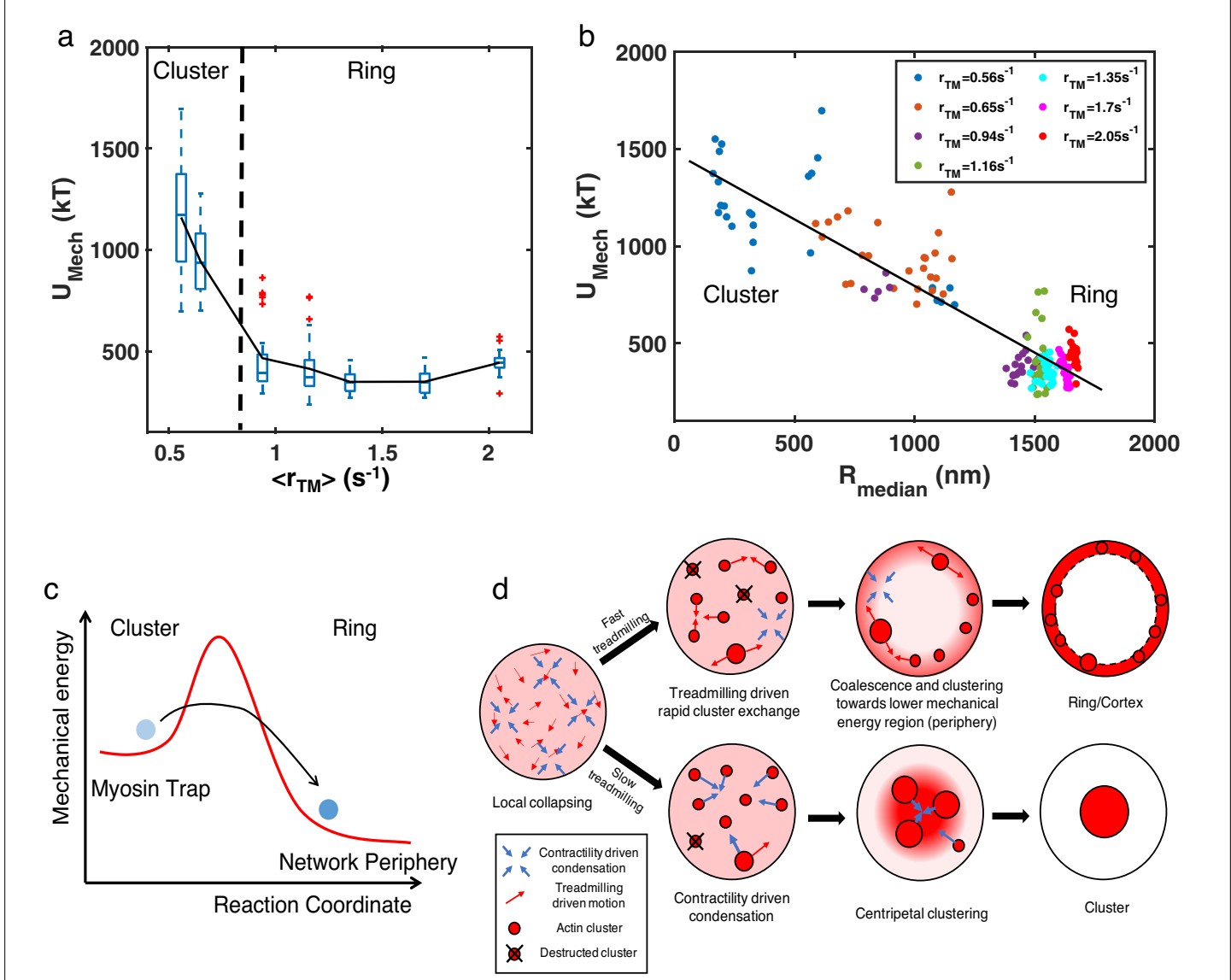

**Figure 6.** Energetic origins of actin rings. (**a**) The box plot shows the steady state $U_{Mech}$ at each treadmilling rate. $U_{Mech}$ is the sum of the bending energy of actin filaments and the stretching energy of filaments, motors, and linkers. The solid line connects the mean $U_{Mech}$ at each $\langle r_{TM} \rangle$. (**b**) Mechanical energy ($U_{Mech}$) and the corresponding $R_{median}$ at different $\langle r_{TM} \rangle$ as indicated in the legend. Each data point represents the average $U_{Mech}$ and $R_{median}$ per 100 s of the last 500 s of simulation. (**a–b**) $C_{actin} = 40\mu M$, $C_{NMII} = 0.06\mu M$, $C_{alpha-actinin} = 4\mu M$, with varying $\langle r_{TM} \rangle$ as shown in **Figure 3a–c**. n=5 runs per condition. (**c**) A graphical description showing the proposed energy landscape for generating actin cortices. (**d**) Schematic showing the formation of actin ring/cortex *versus* clusters. At low treadmilling rates, networks are dominated by myosin-driven contraction, leading to centripetal collapse into clusters (lower). Faster filament treadmilling allows networks to overcome the myosin-driven centripetal motion, where filaments tend to move to the network periphery due to lower energy (upper).

The online version of this article includes the following figure supplement(s) for figure 6:

**Figure supplement 1.** Soluble molecules show no spatial dependence in clustered or ring-like networks.

The lack of enrichment of soluble molecules in the periphery suggested a possible energetic origin of the structures. We thus examined the mechanical energy ($U_{Mech}$) of the system, which primarily arises from filament bending in our simulations. For fixed concentrations of NMII (0.06μM) and cross-linker (4 μM), we found that $U_{Mech}$ decreases with increasing $\langle r_{TM} \rangle$ (**Figure 6a**). In addition, $U_{Mech}$ undergoes a sharp reduction when $\langle r_{TM} \rangle$ reaches the critical threshold, with $U_{Mech}$ of actin rings being two- to threefold lower than that of clusters. Moreover, we found that $U_{Mech}$ is negatively correlated with $R_{median}$, regardless of the structural state (**Figure 6b**). Since higher $R_{median}$ indicates localization

of actin filaments at the network periphery, this negative correlation indicates that configurations with the lowest mechanical energy are those with a ring-like geometry. These results suggest that the peripheral arrangement of actin filaments is more energetically favorable than more distorted configurations found in centripetal clusters.

## Discussion

Detailed mechanochemical modeling using MEDYAN shows that active actin networks exhibit a striking morphological transition upon changes in the filament treadmilling rate. We found that two distinct types of dynamic structures emerge due to the interplay between treadmilling rates and NMII contractility in an initially disordered network: (1) actin clusters formed in slow-treadmilling or high $C_{NMII}$ networks and (2) ring-like and cortex-like structures spontaneously assembled in fast-treadmilling and low $C_{NMII}$ networks. This geometric transition does not require filament tethering to the boundary or spatially biased filament assembly. We also observed a sharp transition in the system's mechanical energy during the transformation from a multi-cluster network to a ring architecture. Such a sharp change in morphology and mechanical energy, induced by tuning filament treadmilling speed, is indicative of a finite size phase transition.

While phase transitions in many biomolecular systems are often driven by passive biomolecular interactions (*Li et al., 2012*; *Brangwynne et al., 2009*), in this work we identified a phase transition in cytoskeletal networks that is induced by non-equilibrium actomyosin dynamics. Our analysis shows that the formation of actin rings and cortices arises from the competition between filament treadmilling and myosin induced contraction. The addition of myosin motors and crosslinkers to an initially disordered actin network induces contractile forces, creating actomyosin clusters having higher mechanical energy (*Figure 6c*). In the language of dissipative structures (*Glansdorff et al., 1973*), actomyosin clusters are thereby trapped in a non-equilibrium, metastable state that cannot easily transition to a final steady state structure with lower mechanical energy. Rapid filament treadmilling provides a mechanism for escaping these traps (*Kim et al., 2014*; *Popov et al., 2016*; *McCall et al., 2019*), giving rise to smaller clusters that rapidly dissolve and reappear. In this state, the network has more freedom to remodel its structure in order to lower the mechanical energy. Indeed, our analysis suggests that as the actin filament distribution shifts to the network periphery, the smaller curvature at the boundary results in a decrease in filament bending, thereby lowering the mechanical energy of the network (*Figure 6a–b*). As a consequence, actin filaments at high treadmilling speeds rapidly accumulate at the network periphery, contributing to the build up of an actin ring in flattened volumes or actin cortices in fully 3D spherical geometries (*Figure 6d*, upper). In contrast, networks undergoing slow filament treadmilling are trapped in cluster-like configurations that have higher mechanical energy. The latter networks are dominated by myosin-driven contractility, leading to a highly non-ergodic state in which actin filaments undergo centripetal collapse (*Figure 6d*, lower).

Although some other modeling studies have studied how network morphology and contractility are regulated by treadmilling rates or stochastic motion of actin filaments (*Vavylonis et al., 2008*; *Kim et al., 2014*; *Mak et al., 2016*; *Oelz et al., 2015*), the formation of ring-like or cortical shell-like networks under active force and their underlying mechanisms have not been examined before. In this work, we examined the impact of actin filament treadmilling and myosin contractility on actin structure using a computational model and validated our findings in experiments. Although results from the simulations are quantitatively in agreement with experiments, we note some of the limitations of our model and suggest future directions to improve our simulations. First, we note that we did not explicitly include some significant properties of actin networks in vivo due to the prohibitively high computational overhead associated with modeling Arp2/3-mediated branching and steric interactions. Second, while we used concentrations of cytoskeletal proteins and their spatial distribution that are largely in agreement with the literature, precise measurements of these will significantly improve the simulations.

Since both the ring state and the cluster state conceptually have lower structural entropy compared to the uniform disordered state, we believe that the driving force for actin ring formation is energetic in origin. However, additional work is needed to quantitatively estimate the entropic contribution to actomyosin network self-organization to further validate this argument. The contribution of filament orientation and length to the formation of ring structure remains to to be determined. Some studies have observed that the assembly of ring-like cytoskeletal structures can be achieved by generating

long filaments that are mechanically compressed by confinement, or by tethering filaments to the network boundary or membrane (*Miyazaki et al., 2015*; *Dmitrieff et al., 2017*; *Nguyen et al., 2018*; *Adeli Koudehi et al., 2019*; *Litschel et al., 2021*). We have shown that forming long filaments is not necessary for generating actin rings, however, filament length can still be an critical parameter in modulating actin network morphology and should be explored in the future. Furthermore, it is likely that filament binding to the cell membrane (*Litschel et al., 2021*) or the spatially biased localization of actin assembly regulators, such as Arp2/3 (*Murugesan et al., 2016*), can further enhance the formation of ring-like structures.

In summary, we have shown that rapid treadmilling and the presence of myosin are sufficient to create ring-like or cortex-like actomyosin networks in a system with confined boundaries. These observations suggest that T cells may modulate the actin treadmilling speed or myosin activity upon stimulation by antigen-presenting cells, which generates the actin ring, which is a hallmark of the immunological synapse. On the other hand, cell types that do not assemble ring-like or shell-like actin structures may have intrinsically slower filament treadmilling or higher myosin contractility. Studying these and other regulatory processes will bring new mechanistic insights into the organization and dynamics of cortices/rings and their defects, which occur in primary immunodeficiencies, autoimmune disorders, and cancers.

## Materials and methods

### Cell culture and transfection

E6.1 Jurkat T cells (a gift from Brian C. Schaefer, Uniformed Services University, MD, USA) were grown in RPMI medium supplemented with 10% Fetal Bovine Serum (FBS) and 1% penicillin-streptomycin at 37°C in a $CO_2$ incubator. Transfections were performed with $2 \times 10^5$ cells using 1 μg of plasmid by electroporation using a Neon electroporation kit (Thermo Fisher Scientific). Prior to imaging, cells were transferred to $CO_2$ independent L-15 medium (Fisher Scientific). Cells tested negative for mycoplasma contamination using MycoAlert Mycoplasma Detection Kit (Lonza).

### Plasmids and reagents

pEGFP-C1 F-tractin-EGFP was a gift from Dyche Mullins (Addgene plasmid # 58473) (*Belin et al., 2014*). The tdTomato-F-tractin plasmid was a gift from Dr. John A. Hammer and the MLC-EGFP plasmid was a gift from Dr. Robert Fischer, National Heart, Lung, and Blood Institute. Latrunculin A was purchased from Sigma Aldrich Calyculin A was purchased from Cell Signaling Technology, Y-27632 was purchased from Selleck Chemicals, and dimethyl sulfoxide (DMSO) was purchased from Thermo Fisher Scientific.

### Preparation of glass coverslips

Sterile eight-well chambers (Cellvis) were incubated with 0.01% poly-L-lysine solution in distilled water for 10 min and then dried at 37°C for 1hr. Poly-L-lysine-coated chambers were then incubated with anti-human CD3 antibody (HIT3a clone, Thermo Fisher Scientific) in PBS at a concentration of 10 μg/mL for 2 hr at 37°C or overnight at 4°C. Following incubation, the chambers were washed five times with L-15 and warmed prior to imaging.

### Microscopy

Transfected T cells were seeded on anti-CD3 coated glass coverslips and allowed to activate for 5 min. Chambers were maintained at 37°C using a stage-top incubator (Okolab). Latrunculin A or vehicle (DMSO) were added at specified concentrations 5 min after seeding the cells. Fluorescence and interference reflection microscopy (IRM) images were acquired using an inverted microscope (Ti-E, Nikon, Melville, NY) with a scientific CMOS camera (Prime BSI, Photometrics, Tucson, AZ) with a frame interval of 2s. F-tractin-EGFP was imaged using total internal reflection fluorescence (TIRF), using a 60X, 1.49 NA oil immersion objective. One background image was captured during every session in order to perform background subtraction.

For inhibitor experiments with Calcyulin-A and Y-27632, 50 nM Calcyulin-A, 100 μM Y-27632 or vehicle (DMSO) were added after the cells had formed an actin ring. TIRF images were acquired as above with a frame interval of 2 s using a 100X, 1.49 NA oil immersion objective.

## Image analysis

Initial preprocessing of images was done using Fiji (*Schindelin et al., 2012*). A custom MATLAB script was written to perform background subtraction. The IRM or actin images were used to find the outline and centroid of the cells. 50 uniformly spaced lines were drawn from the centroid and these 50 line profiles were pooled together to generate a histogram of intensities as a function of a normalized distance to the centroid. The median of the distribution of intensities (and hence F-actin) was estimated for each time point. Custom MATLAB script can be found in the repository in the Data Availability Statement.

To calculate the slope of center to plateau F-actin distribution, a cell mask was drawn for each cell (*Figure 5—figure supplement 1*, left - yellow outline) using a minimum threshold intensity. The centroid (red dot) of the masked cell was identified, and 50 equally spaced lines joining the centroid to the mask edge were drawn and the intensity profile was averaged over all these lines. This plot gives a single intensity line profile from cell centroid to cell edge for a cell at a given time point. Similarly, the line profiles for all the other time points spaced 30 s apart are obtained and normalized using the mean intensity of the cell to account for the effects of photobleaching. The resultant normalized line profile curves are now representative of how the actin distribution changes over time inside the cell (*Figure 5—figure supplement 1*, right). The intensity profiles typically display a linear regime before they plateau near the cell edge. The linear region of the line profile curves are fit to straight lines (*Figure 5—figure supplement 1*, right - shaded red and blue lines) to find their slope at each time point and the changes in the slopes over time are then compared across different chemical perturbations.

## Simulation methods

### Simulation setup overview

In this work, we employed an open-access mechanochemical platform for simulating active matter (MEDYAN *Popov et al., 2016*) to investigate the spatiotemporal evolution of actin networks under different treadmilling and myosin motor conditions. MEDYAN accounts for two overlapping phases and their interactions. (1) Diffusing G-actin and unbound formins, NMII and linkers are spatially dissolved in a solution phase. In this phase, the network is discretized into compartments based on the Kuramoto length of G-actin, which is the mean-free path that G-actin molecules are expected to diffuse before undergoing their next reaction (*Hu and Papoian, 2010*). Diffusing chemical species are assumed to be well-mixed within each compartment, and inter-compartment transports are modeled as stochastic diffusion reactions. (2) Polymeric filaments and bound species comprise the continuous polymeric phase which is overlaid on the solution phase. The polymeric phase is mechanically active, where filament bending, stretching, and steric interactions are taken into account. Bound motors and linkers are modeled as harmonic springs based on the mechanical properties of NMII and alpha-actinin. A boundary repulsion potential restricts filaments within the volume boundary. Filament polymerization is affected by interactions with the boundary, following the Brownian Ratchet model (*Peskin et al., 1993*). The following chemical reactions stochastically occur among the two phases: filaments can polymerize, depolymerize, and interact with myosin and crosslinker; formins are able to bind to G-actin and nucleate filaments; filaments that are only two monomers long can be rapidly destroyed. The chemical reaction modeling engine is based on an efficient and statistically accurate Next Reaction Method (NRM) (*Gibson and Bruck, 2000*), which is a variant of the Gillespie Algorithm (*Gillespie, 1977*).

We initialized de novo cytoskeletal networks in MEDYAN with small seed filaments, 40 μM diffusing G-actin, and 100 nM filament nucleators based on their reported cytoplasmic concentrations in cells (*Wu and Pollard, 2005*; *Kiuchi et al., 2011*; *Dominguez and Holmes, 2011*). Most of the simulations were carried out in a thin oblate geometry, having a diameter ranging from 3.8 μm to 10 μm and an effective height of 200 nm. The spherical simulation volume has a diameter of 4 μm. We tuned the barbed end polymerization rate and pointed end depolymerization rate to model the effects of treadmilling promoters such as formin, profilin, and cofilin. To monitor the actual speed of treadmilling, we define $\langle r_{TM} \rangle$ as the average barbed end elongation rate, which is also equal to the shortening rate of the pointed end at steady state. Networks were allowed to assemble with only filament polymerization, depolymerization, nucleation, and disassembly for 300 s. At 300 s, 0.06 μM NMII and 4 μM

alpha-actinin crosslinkers are added. The local density of clusters and rings were measured using a customized density based clustering algorithm.

## Mechanical models

Unlike the traditional bead-spring model, the semi-flexible filaments are represented as connected cylinders. The equilibrium length (under zero force) of each cylinder elements varies from 2.7 nm (1 actin monomer) to a maximum of 108 nm (40 actin monomers). Addition of each actin monomer would increase the length of the first or last cylinders by 2.7 nm, and vice versa. Polymerization will create a new cylinder if the cylinder has reached its maximum length. Filaments have a very large aspect ratio, that is, the persistence length of a filament ($\sim 20\mu m$) is much larger than its diameter ($\sim 10nm$). Thus, it is reasonable to ignore the radial stretching/compression and only allow the axial stretching/compression of a cylinder, which is written as

$$U_{filament}^{str} = \tfrac{1}{2}K_{filament}^{str}(l_f - l_{f,0})^2.$$

$l_f$ is the actual length of cylinder under force, and $l_{f,0}$ is the equilibrium length based on the number of actin monomers on this cylinder (each monomer is 2.7 nm). Radial filament deformation is modeled as bending between two connected cylinders:

$$U_{filament}^{bending} = K_{filament}^{bending}(1 - cos(\theta - \theta_0)),$$

where $\theta$ is the angle between the two consecutive cylinders under force, while $\theta_0$ is the equilibrium angle that is set to be 0.

A novel volume exclusion exclusion potential is implemented to prevent cylinders overlapping, which is written as

$$U^{Vol} = \iint_{l_i,l_j} \delta U \mid \vec{r}_i - \vec{r}_j \mid dl_i dl_j,$$

where $\delta U \mid \vec{r}_i - \vec{r}_j \mid= 1/\mid \vec{r}_i - \vec{r}_j \mid^4$ is the pair potential between two points located on the two interacting cylinders. $\vec{r}_i$ amd $\vec{r}_j$ are the distances between any two points along the cylinder $i$ and $j$, respectively. This potential can provide a steep enough volume exclusion effect while remain analytically solvable. Bound NMIIs and linkers are modeled as harmonic springs, and the stretching energy is written as

$$U_{NMII/linker}^{str} = \tfrac{1}{2}K_{NMII/linker}^{str}(l_{NMII/linker} - l_{NMII/linker,0})^2.$$

$l_{NMII/linker,0}$ is the equilibrium length of a linker, which are initialized when a linker /NMII binding reaction occurs as the distance between the paired binding site. $l_{NMII,0}$ is reset every time a motor walking reaction occurs.

In order to confine all the filaments within the simulation boundary, an exponential boundary repulsion potential is implemented. In the thin oblate system, the actual height of the network is set to be 400nm, and the diameter to 4000 nm. However, filaments would occasionally move out of the mechanical boundary due to rapid treadmilling, leading to simulation failures. To prevent this, we

**Table 1.** Mechanical parameters.

| Names | Parameters | References |
|---|---|---|
| Cylinder stretching | $K_{filament}^{str} = 100pN/nm$ | *Popov et al., 2016* |
| Cylinder bending | $K_{filament}^{bending} = 672pN \cdot nm$ | *Ott et al., 1993* |
| Filament volume exclusion | $K_{vol} = 10^5 pN/nm^4$ | *Popov et al., 2016* |
| Linker stretching | $K_{linker}^{str} = 8pN/nm$ | *DiDonna and Levine, 2007* |
| NMII stretching | $K_{NMII}^{str} = 2.5pN/nm$ per head | *Vilfan and Duke, 2003* |
| Boundary repulsion | $\epsilon_{boundary} = 100pN \cdot nm$ | This work |

shift the boundary barrier slightly inside the network by $a_0$, and the exponential boundary repulsion is written as

$$U^{boundary} = \epsilon_{boundary} e^{-(d-a_0)/\lambda},$$

where $\epsilon_{boundary} = 100 pN \cdot nm$ is the repulsive energy constant, $d$ is the distance between boundary and filament element, and $\lambda = 2.7nm$ is the screening length. The boundary shifting factor $a_0$ is chosen to be 100nm based on experience. The existence of $a_0$ restricts the effective network boundary to height =200 nm and diameter =3800 nm.

The mechanical model parameters can be found in *Table 1*.

## Chemical models

The chemical engine of MEDYAN is powered by Next Reaction Method (NRM)(*Gibson and Bruck, 2000*), which is a variant of the Gillespie algorithm (*Gillespie, 1977*). Overall, the NRM stochastically solves the chemical Master Equation by generating a trajectory of chemical events. In this work, we simulated the following chemical reactions: diffusion, filament polymerization, filament depolymerization, filament nucleation, destruction of filaments, binding of myosin motors and linkers, and motor walking.

The diffusion of molecules is modeled as a single molecule transfer process between neighboring compartments, which follows our stochastic chemical reaction protocol as

$$DM_{i_0,j_0,k_0} \longrightarrow DM_{i_1,j_1,k_1},$$

where a diffusing molecule (DM) originally located in compartment $_{i_0,j_0,k_0}$ is transferred to a neighboring compartment $_{i_1,j_1,k_1}$. The copy number of this diffusing molecule species is decreased by 1 in compartment $_{i_0,j_0,k_0}$ and is increased by 1 in compartment $_{i_1,j_1,k_1}$.

Actin filament (F-actin) polymerization and depolymerization occur at both barbed end (BE) and pointed end (PE) of a filament. These reactions are written as

$$G - actin \longrightarrow F - actin,$$
$$F - actin \longrightarrow G - actin$$

It should be noted that G-actin is dissolved in the solution phase, while F-actin is in the polymeric phase.

The nucleation reaction is presented as a two-step reaction based on the mechanism of formin nucleation (*Pring et al., 2003*; *Ni and Papoian, 2019*):

$$Step\ 1 : Formin + G - actin \longrightarrow intermediate,$$
$$Step\ 2 : G - actin + intermediate \longrightarrow FBE - actin + F - actin + PE - actin.$$

The intermediate is an arbitrary molecule that consists of a formin and a G-actin molecule. We assume step 1 is the rate-limiting step and step 2 is a fast step, thus this intermediate would rapidly react with a G-actin molecule and become a short filament consisting of one F-actin molecule at the pointed end (PE-actin), a regular F-actin molecule, and another F-actin molecule at the formin bound barbed end (FBE-actin). For simplicity, polymerization and depolymerization at FBE are the same as regular barbed end reactions. Formin can dissociate from a filament, which releases a formin molecule into the solution phase and creates a regular F-actin barbed end (BE-actin) on that filament:

$$F - actin + FBE - actin \longrightarrow BE - actin + Formin.$$

Since new filaments are constantly created by nucleation,, the filament destruction process is required to establish a steady state which maintains a constant total number of filaments. The destruction reaction occurs exclusively when a filament has only two F-actin molecules (a BE-actin and a PE-actin), which destroys this filament and releases two diffusing G-actin molecules as

$$BE - actin + PE - actin \longrightarrow 2G - actin.$$

The binding reactions of myosin motors and linkers are carried out with a slightly different protocol. Firstly, the system will search for all possible binding site pairs on actin filaments and stochastically

**Table 2.** Parameters for diffusion and reactions.

| Names | Parameters | References |
|---|---|---|
| Diffusion | $D_{actin,arp2/3,CP} = 20\mu M^2/s$ | *Hu and Papoian, 2010* |
| Actin | $k_{on}^{BE} = 11.6 - 34.8(\mu M \cdot s)^{-1}$ | 32 and this work |
| | $k_{on}^{PE} = 1.3(\mu M \cdot s)^{-1}$ | |
| | $k_{off}^{BE} = 1.4s^{-1}$ | |
| | $k_{off}^{PE} = 0.8 - 2.4s^{-1}$ | |
| Destruction | $k_{destruction} = 1.0 - 1.9s^{-1}$ | This work |
| Nucleation | $k_{nu} = 0.005s^{-1}$ | *Ni and Papoian, 2019* |
| Formin dissociation | $k_{off}^{formin} = 0.01s^{-1}$ | *Fritzsche et al., 2016* |
| Alpha-actinin | $k_{on}^{\alpha} = 0.7(\mu M \cdot s)^{-1}$ | *Wachsstock et al., 1993* |
| | $k_{on}^{\alpha} = 0.3s^{-1}$ | |
| NMII head binding | $k_{on}^{M} = 0.2s^{-1}$ | *Kovács et al., 2003* |
| | $k_{on}^{M} = 1.7s^{-1}$ | *Popov et al., 2016* |

choose one for binding reaction. The two binding sites of a pair must be located at different filaments. The distance between the two binding sites ranges from 175 to 225 nm for NMII mini filament (*Pollard, 1982*), and 30–40 nm for alpha-actinin crosslinker (*Meyer and Aebi, 1990*). After the binding site pair is determined, the binding reaction convert a diffusing motor or linker to a bound motor or linker with two ends attaching to the two binding sites, creating a mechanical linkage. This linkage vanishes when an unbinding reaction occurs, releasing the motor or linker to the diffusing pool. It should be noted that NMII mini filament is an ensemble of 15–30 myosin heads (*Verkhovsky et al., 1995*), and we model the entire ensemble as a while. To take the variation of the number of myosin heads into account, the number of myosin heads of each NMII mini filament is chosen stochastically for each reaction, and the reaction rate for each NMII binding event is then scaled by the number of myosin heads.

In an active cytoskeleton, myosin motors consume energy from ATP hydrolysis and actively walk along filaments, which is one of the most important sources of contractile force generation. In MEDYAN, a motor stepping reaction is implemented to mimic this effect. For a bound NMII, the stepping reaction is written as

$$NMII_i \longrightarrow NMII_{i+1},$$

where $i$ and $i + 1$ are the NMII locations on the filament before and after walking. NMII is a barbed end walking motor, thus $i + 1$ represents the next binding site towards the barbed end.

Parameters for diffusion and chemical reactions can be found in *Table 2*.

## Mechanochemical models

Many cytoskeletal reactions, including actin polymerization, myosin motor binding and stepping, and linker binding, are mechanosensitive. To capture this feature, MEDYAN implements mechanochemical models that explicitly allow force-dependent chemical reaction rates.

The effect of boundary force on filament polymerization is described by the Brownian Ratchet model (*Peskin et al., 1993*), which models the force sensitive polymerization rate $k_{poly}$ as:

$$k_{poly} = k_{poly}^0 \cdot \exp(-F_{ext}/F_{poly,0}),$$

**Table 3.** Mechanochemical dynamic rate parameters.

| Names | Parameters | References |
|---|---|---|
| Characteristic polymerization force | $F_{poly,0} = 1.5pN$ | *Footer et al., 2007* |
| Characteristic linker unbinding force | $F_{linker,unbind} = 17.2pN$ | *Ferrer et al., 2008* |
| NMII duty ratio | $\rho = 0.1$ | *Kovács et al., 2003* |
| NMII stall force | $F_{stall} = 12.62pN$ per head | *Erdmann et al., 2013* |
| Tunable parameters | $\beta = 0.2$ | *Popov et al., 2016* |
| | $\gamma = 0.05pN^{-1}$ | |
| | $\xi = 0.1$ | |

where $k_{poly}^0$ is the bare polymerization rate under zero external force, $F_{ext}$ is the boundary repulsive force exerted on the filament ends, and $F_{poly,0}$ is the characteristic polymerization force based on the thermal energy and the size of actin monomers.

We used a simple exponential equation to model the slip bond property of alpha-actinin crosslinker:

$$k_{linker,unbind} = k_{linker,unbind}^0 \cdot \exp(F_{linker,stretching}/F_{linker,unbind}),$$

where $k_{linker,unbind}^0$ is the unbinding rate constant under zero external force, and $F_{linker,unbind}$ is the characteristic unbinding force of alpha-actinin. $F_{linker,stretching}$ is the stretching force on the linker, while a compressive force on the linker does not trigger the slip bond.

In this work, we model NMII binding as a catch bond, as adapted from the Parallel Cluster Model (*Erdmann et al., 2013*), such that the force loaded on NMII can reduce its unbinding rate constant:

$$k_{NMII,unbind} = \frac{\beta \cdot k_{NMII,unbind}^0}{N_{heads}} \cdot exp(\frac{-F_{ext}}{N_{heads} \cdot F_{NMII,unbind}}),$$

where $\beta$ is a tunable parameter, $k_{NMII,unbind}^0$ is the unbinding rate constant under zero force, $F_{ext}$ is the total stretching force applied on the NMII, and $N_{heads}$ is the number of NMII heads.

The NMII walking rate is also mechanochemically sensitive and can be modeled with a Hill type force-velocity relation:

$$k_{walk} = k_{walk}^0 \cdot \frac{F_{stall} - F_{ext}/N_{heads}}{F_{stall} + F_{NMII,pulling}/(N_{heads} \cdot \xi)},$$

where $F_{stall}$ is the stall force of a single NMII head, $F_{NMII,pulling}$ is the pulling force on NMII in the opposite direction of walking movement, and $\xi$ is a tunable parameter.

The mechanochemical model parameters can be found in *Table 3*.

## Simulation protocol

The relaxation time for local deformations of actin networks (*Falzone et al., 2015*) is much shorter than the timescale of typical chemical events such as motor stepping (*Kovács et al., 2003*) or filament polymerization (*Fujiwara et al., 2007*), thereby creating a significant separation of timescales. Hence, the mechanical equilibrium process can be viewed as a pseudo-adiabatic process that can be separated from chemical reactions. Based on this hypothesis, the simulation can be carried out in the following steps:

1. Chemical reactions occur that evolve the time of the system stochastically.
2. Pausing chemical reactions when the time step reaches a preset value, which is 10ms in this work. The system then mechanically minimizes the total energy.
3. Reaction rates are updated based on the tension acting on NMIIs/linkers and load force acting on actin filament barbed ends after mechanical minimization.
4. Step 1 is repeated based on the updated reaction rates.

This protocol is iterated until we reach 2000 s of simulation time, or until we reach the wall time limit on the Deepthought2 High-Performance Computing cluster at University of Maryland, College Park, whichever comes first.

## Defining treadmilling rate and treadmilling inhibition simulation setups

Although treadmilling in cells is a complex system that involves hundreds of reactions (*Bugyi and Carlier, 2010*; *Floyd et al., 2017*), it is simplified to four reactions in this work by considering polymerization and depolymerization at both barbed ends and pointed ends. When a steady state is established, the net barbed end growth rate will equal the net pointed ends reduction rate (averaged over the system), maintaining a constant average filament length. Therefore, we can define a kinetic steady state for treadmilling by monitoring the average filament length of the network as shown in *Figure 2—figure supplement 7*. We found that such a kinetic steady state could be established after 1000 s in all conditions, and at this state, the average barbed end elongation rate is almost the same as the average pointed end shrinkage rate. Hence, we quantify the average treadmilling rate $\langle r_{TM} \rangle$ as the average barbed end elongation rate after 1000 s.

While the treadmilling rate is an elegant and robust way of quantifying the speed of actin network assembly, it is extremely hard to measure in vivo. An alternative way to quantify the speed of actin network remodeling is to measure the turnover timescale, which has been widely studied via an experimental technique called Fluorescent Recovery After Photobleaching (FRAP). To compare with experiments, in our simulation we used a method mimicking the FRAP to calculate the turnover halftime ($t_{1/2}$, the time required for a network to reach 50% turnover) as developed in our previous work (*Ni and Papoian, 2019*), and we obtain $t_{1/2} \sim 168s$ for the slowest treadmilling condition, and $t_{1/2} \sim 48s$ for the most rapid treadmilling case. It should be noted that our longest $t_{1/2}$ is similar to the turnover timescale of some reconstituted networks (*McCall et al., 2019*), and our shortest $t_{1/2}$ is comparable to that of in vivo actin cortices (*Salbreux et al., 2012*). The details of turnover halftime measurement in MEDYAN and how it is related to treadmilling has been discussed in depth in a prior computational study (*Ni and Papoian, 2019*).

We utilized kinetic parameters measured in vitro (*Fujiwara et al., 2007*) as the baseline to assemble the slow treadmilling networks. To explore suitable parameters for rapidly treadmilling networks, we looked into the effects of formin and ADF/cofilin. An earlier work (*Kovar et al., 2006*) has shown that the presence of formin can boost the polymerization rate at the barbed end several-fold over the baseline. For simplicity, we imitated this effect by increasing the barbed end polymerization rate constant ($k_{on}^{BE}$). ADF/cofilin can also promote treadmilling by severing filaments. Importantly, the fragment that contains the pre-existing pointed end is very unstable and would undergoes rapid disassembly (*McCall et al., 2019*). This observation allows us to mimic the effect of ADF/cofilin by simply increasing the depolymerization at the pointed end ($k_{off}^{PE}$). For example, we increase the $k_{on}^{BE}$ and $k_{off}^{PE}$ to three-fold in the actin ring network as shown in *Figure 1a–c* ($\langle r_{TM} \rangle = 2.05s^{-1}$).

## Calculation of local actin concentration for clusters and rings

In this work, we used a density-based clustering method to define regions that contain actin clusters and rings, and calculated the local F-actin concentration within these regions. We first a generated pixelated map by dividing the network into $100nm \times 100nm$ bins and calculated the F-actin concentration within each bins (*Figure 2—figure supplement 2a*). We then grouped connecting bins with concentration higher than a threshold (160 μM) into clusters (*Figure 2—figure supplement 2b*). Clusters with size less than 4 bins were ignored. The local actin concentration within clusters was calculated as the average F-actin concentration of these clusters. The local actin concentration within actin rings is calculated using the same method (*Figure 2—figure supplement 2c-d*).

## Simulation setups of Latrunculin A, Calyculin A, and Y-27632 modeling

Earlier works have shown that LatA affects filament treadmilling in two ways: (1) it sequesters G-actin and (2) it accelerates the phosphate release from ADP-Pi-actin thereby reducing filament polymerization while increasing depolymerization at both ends (*Lodish, 2000*; *Yarmola et al., 2000*; *Fujiwara et al., 2018*). To simulate such effects in the actin ring perturbation simulations, we explore a parameter space that mimicked the effect of LatA treatment: we disrupted $r_{TM}$ by reducing the filament polymerization rate and increasing the depolymerization rates. In the weak inhibition case,

we decreased $k_{on}^{BE}$ to $11.6(\mu M \cdot s)^{-1}$, increased $k_{off}^{BE}$ to $2.1 s^{-1}$, and maintained $k_{off}^{PE}$ at $2.4 s^{-1}$. In the strong inhibition case, $k_{on}^{BE}$ was decreased to $3.48(\mu M \cdot s)^{-1}$, $k_{off}^{BE}$ was increased to $11.2 s^{-1}$, and $k_{off}^{PE}$ was increased to $4.8 s^{-1}$. In all simulations, pointed end polymerization rate was set to be constant at $1.3(\mu M \cdot s)^{-1}$. Treadmilling rate is consequentially reduced as a result of such disruption.

Calyculin A is an enhancer of NMII activity by inhibiting myosin light chain ATPase, while Y-27632 inhibits Rho kinase, a upstream regulator of NMII. Thus, we model their effects by increasing or decreasing the NMII levels after actin ring formation to match the T cell experiment. In the CalyA experiment, actomyosin ring collapses while maintaining the ring-like geometry. We realize that such 'whole ring contraction' is difficult to achieve at the low actin concentration ($C_{actin} = 40 \mu M$) that we used other conditions. At low actin concentration, enhancing NMII activity often simultaneously cause centripetal collapse as well as the local collapse that disassemble the ring-like structure, due to lack of filament-filament connectivity. To overcome this issue, we double the actin concentration to $C_{actin} = 80 \mu M$ and adjust $C_{NMII}$ to 0.18 µM in the model. Such a high concentration of actin and motor protein significantly reduces the computational efficiency, therefore we initialize the ring-like actin structure instead of starting from a disordered network. In the control condition as shown in Figure S9, network will slightly contract but can maintain the ring-like structure.

## Acknowledgements

We thank A Chandrasekaran, C Floyd, and J Komianos for helpful discussions and feedback on the manuscript. This work was supported by National Science Foundation grants CHE-2102684 and PHY-1806903. AU acknowledges support from NSF grant PHY 1607645 and NIH grant R35 GM145313. Computational resources were provided by Deepthought2 HPC at University of Maryland.

## Additional information

### Funding

| Funder | Grant reference number | Author |
|---|---|---|
| National Science Foundation | CHE-2102684 | Garegin A Papoian |
| National Science Foundation | PHY-1806903 | Garegin A Papoian |
| National Science Foundation | PHY-1607645 | Arpita Upadhyaya |
| National Institutes of Health | R35 GM145313 | Arpita Upadhyaya |

The funders had no role in study design, data collection and interpretation, or the decision to submit the work for publication.

### Author contributions

Qin Ni, Conceptualization, Data curation, Software, Formal analysis, Investigation, Visualization, Methodology, Writing – original draft, Project administration, Writing – review and editing; Kaustubh Wagh, Conceptualization, Data curation, Investigation, Visualization, Methodology, Writing – original draft, Writing – review and editing; Aashli Pathni, Formal analysis, Investigation, Methodology, Writing – review and editing; Haoran Ni, Software, Investigation, Methodology, Writing – review and editing; Vishavdeep Vashisht, Formal analysis, Investigation, Visualization, Writing – review and editing; Arpita Upadhyaya, Conceptualization, Supervision, Funding acquisition, Methodology, Writing – original draft, Project administration, Writing – review and editing; Garegin A Papoian, Conceptualization, Software, Supervision, Funding acquisition, Investigation, Methodology, Writing – original draft, Project administration, Writing – review and editing

### Author ORCIDs

Qin Ni (iD) http://orcid.org/0000-0002-0738-1817

Kaustubh Wagh http://orcid.org/0000-0001-8514-027X
Aashli Pathni http://orcid.org/0000-0003-4196-890X
Vishavdeep Vashisht http://orcid.org/0000-0002-9367-0278
Arpita Upadhyaya http://orcid.org/0000-0003-1496-919X
Garegin A Papoian http://orcid.org/0000-0001-8580-3790

Decision letter and Author response
Decision letter https://doi.org/10.7554/eLife.82658.sa1
Author response https://doi.org/10.7554/eLife.82658.sa2

## Additional files

### Supplementary files
• MDAR checklist

### Data availability
Source Data files for experiments and the modeling code are available in Digital Repository at the University of Maryland(DRUM): https://doi.org/10.13016/9t26-ovid.

The following dataset was generated:

| Author(s) | Year | Dataset title | Dataset URL | Database and Identifier |
|---|---|---|---|---|
| Ni Q, Wagh K, Pathni A, Ni H, Vashisht V, Upadhyaya A, Papoian GA | 2022 | Data for "A tug of war between filament treadmilling and myosin induced contractility generates actin ring" | https://doi.org/10.13016/9t26-ovid | Digital Repository at the University of Maryland, 10.13016/9t26-ovid |

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
