## [Editor Report]

This important paper uses molecular simulations to explain how actomyosin networks transition from small clusters to the cortex or ring-shaped actin networks. The authors provide compelling evidence that variation in filament turnover rate and myosin concentration triggers a phase transition of these networks. The predictions of this model are consistent with observations made in T cells, where actin ring formation can be induced following their activation by antibodies.

---

## [Decision Letter]

**Decision letter after peer review:**

Thank you for submitting your article "A tug of war between filament treadmilling and myosin induced contractility generates actin ring" for consideration by *eLife*. Your article has been reviewed by 2 peer reviewers, and the evaluation has been overseen by a Reviewing Editor and Anna Akhmanova as the Senior Editor. The following individual involved in the review of your submission has agreed to reveal their identity: Anatoly B. Kolomeisky (Reviewer #1).

Essential revisions:

You will find that the reviewers have a very positive opinion of your work. Their comments should be easy to address. Please respond to them point by point.

*Reviewer #1 (Recommendations for the authors):*

I have several specific comments and suggestions that might improve the paper:

1) It would be nice to discuss why rings are observed only in a few types of cells, but not in all of them. Is it known?

2) NMII on page 5 should be properly defined first - it is not done here.

3) On page 6, the authors said that they "excluded NMII from the peripheral region..." But what would happen if simulations started with motor proteins uniformly distributed over the system?

4) I would add a brief discussion that entropic terms most probably are not important for this system, and thus the mechanical energy provides a valid thermodynamic quantity to decide about the proper phase. This is because one could naively argue that in the ring structures the entry is reduced due to higher density.

*Reviewer #2 (Recommendations for the authors):*

A] Expanding on public review:

– Additional simulation controls are needed:

While the mean length of filaments according to the depolymerization rate is given, the length distribution in the different conditions is not provided – while this distribution has a strong effect on the ring-like organization of actin in confinement. Moreover, in the simulation of lat A treatment, a control that in-silico actin concentration matches the in-vivo one is missing.

– Confusing description of rings/experiment – simulation discrepancies

For instance, the ring in the control of latA treatment (Figure 4a) seems much more focused than that of WT (Figure 1A). In figure 5, supp. 2 it is hard to understand why the simulated ring is contracting in the control condition. Is time 0 not the stationary solution? Then, how is time 0 chosen?

Also, in figure 1, the ring seems much more realistic than in the following.

I am assuming the authors focused on a simpler system with fewer ingredients to establish the phase diagram, but this needs to be made clearer. Also, why not do a phase diagram with a simpler system, and a more realistic system to compare to experiments?

For now, the comparison between experimental results and simulations is not as strong as claimed. Either the conclusions could be toned down a bit, or the discrepancies should be clearly discussed.

– MEDYAN license

This is not directly relevant to this article but falls under the public review guideline "the utility of the methods and data to the community".

While MEDYAN being an open source software is a great boon for the community, it is crippled by its own license. The item 3 "Users can modify the MEDYAN source code for their own academic and research purposes, but cannot redistribute modified MEDYAN source code that differs in any way from Papoian lab's current MEDYAN distribution" goes against the core idea of open source scientific software: if a team decide to build upon MEDYAN, they will not be able to publish the modified code, and thus will not be able to share their results in a significant manner.

Moreover, the guideline "cannot redistribute any other codes that use or extend any MEDYAN source code" prevents anyone from sharing wrappers and utilities for MEDYAN. Most scientific software uses the GPL (most restrictive) or MIT (most permissive) license with great success.

Our main value as scientists is the production and sharing of knowledge – such a restrictive license does not seem to achieve this goal.

– Patches as a metastable state: this is a bit confusing or over-stated. First, this is a highly out-of-equilibrium active system, and I do not think "metastable" quite applies. Both are fundamentally unstable states. A more likely claim is that treadmilling allows for a faster relaxation of filament elastic energy.

B] Additional comments:

– This is a very well-written article that is highly interesting – yet easy to read.

– The claim that ring organization is not understood, not found in vitro, is far-fetched. First, actin rings in vitro have been observed, e.g. Miyazaki et al. 2015 to name one. Second, simulation of actin rings exists, cf Vavylonys 2008, Hang 2015, Nguyen 2018, Koudehi 2016; see also Dmitrieff 2017 for a ring of microtubules. In all these systems, confinement and filament length plays a major role in a ring formation, but the interplay between the rate of treadmilling and motor activity has not yet been really discussed to my knowledge. This is why not only the mean, but also the distribution of filament length has to be documented, and the role of confinement has to be discussed. The existing literature has to be more discussed.

– The difference in filament orientation in Figure 2 supp 4 is striking. I suspect that this could make for a much more striking phase diagram (by computing some order parameter for instance) than the current phase diagram.

---

## [Author Response]

Reviewer #1 (Recommendations for the authors):I have several specific comments and suggestions that might improve the paper:1) It would be nice to discuss why rings are observed only in a few types of cells, but not in all of them. Is it known?

Why different cell types display distinct higher order structures such as actin rings is not well understood. In some cell types, ring-like actin networks are formed to subserve distinct biological functions. For example, actin rings form in T cells upon stimulation by antigen-presenting cells, where the actin ring dynamically aggregates receptors to initialize the formation of immunological synapses. The assembly of actin ring may also be a consequence of preferential localization of actin regulatory proteins such as Arp2/3 and formins. In neuronal axons, actin organizes into circumferential rings that are evenly spaced along the long axis, providing a structural framework for membrane channel organization. The main purpose of our manuscript is to determine the minimal required conditions for the formation of actin rings. Our results suggest that lack of ring-like structures in some cell types is potentially due to low filament treadmilling rates or high myosin activity, which can be tested experimentally in these systems. We have now highlighted some of these points in the Discussion section (pg, 19-20).

2) NMII on page 5 should be properly defined first - it is not done here.

NMII stands for non-muscle myosin II. We have added this definition to the manuscript.

3) On page 6, the authors said that they "excluded NMII from the peripheral region..." But what would happen if simulations started with motor proteins uniformly distributed over the system?

The Arp2/3 based actin network at the ring periphery forms a dense meshwork1,2, which can sterically exclude NMII mini filaments. Because simulating steric interactions of myosins with actin filaments in dense dendritic networks is computationally expensive (which is why this is not yet currently implemented in MEDYAN), we have taken a simpler approach by excluding NMII from the peripheral region to mimic the steric interactions. If NMIIs are allowed to access all regions in our simulations, we expect that the inner ring and the outer ring will merge. We have elaborated on the reasoning for excluding NMII from peripheral regions in the manuscript (pg. 6).

4) I would add a brief discussion that entropic terms most probably are not important for this system, and thus the mechanical energy provides a valid thermodynamic quantity to decide about the proper phase. This is because one could naively argue that in the ring structures the entry is reduced due to higher density.

We thank the reviewer for bringing up this point. Because the system is far from equilibrium, it is difficult to quantitatively estimate the entropic contribution. Conceptually, the uniform disordered state should have higher structural entropy, while both the ring state and the contracted clusters should have lower entropy. This argument suggests that the driving force for the ring state is energetic in origin. We now mention this in the Discussion section (pg. 20).

Reviewer #2 (Recommendations for the authors):A] Expanding on public review:– Additional simulation controls are needed:While the mean length of filaments according to the depolymerization rate is given, the length distribution in the different conditions is not provided – while this distribution has a strong effect on the ring-like organization of actin in confinement. Moreover, in the simulation of lat A treatment, a control that in-silico actin concentration matches the in-vivo one is missing.

We thank the reviewer for bringing up this point. We have now added a figure showing the filament length distribution at different treadmilling rates (Figure 2-supp. 7b). We find that the filament length (mean length ~ 0.4 um) is much smaller than the confinement size (the diameter of the network is 4um) even in some extreme cases. Thus, we believe filament length is not a critical parameter for ring formation in our simulations. This is different from other in vitro work where long filaments lead to ring formation. We have now added this in the Discussion section of the manuscript (pg. 20).

We note that actin concentration in vivo has not been well quantified, including in T cells. We use 40 µM as the actin concentration, on the same order of 100 µM as reported in other cell types^3–5^. We now explicitly mention the choice of actin concentration in the Simulation Methods section, and have added additional references to address this point (pg. 24).

– Confusing description of rings/experiment – simulation discrepanciesFor instance, the ring in the control of latA treatment (Figure 4a) seems much more focused than that of WT (Figure 1A).

We have replaced Figure 4a with a more representative cell. We note that actin rings in T cells display some degree of heterogeneity in terms of cell size, actin density, ring thickness, etc. Figure 1a shows a clear separation of the inner ring and the outer ring, but in some cases the separation is not as clear. Some of the observed heterogeneity may be also due to differences in the levels of the exogenously expressed fluorescently labeled F-tractin.

In figure 5, supp. 2 it is hard to understand why the simulated ring is contracting in the control condition. Is time 0 not the stationary solution? Then, how is time 0 chosen?

In Figure 5e,f and Supp. Figure 2, time 0 is the timepoint where we start to add or remove NMII, while time 0 in the experiment is when the drug was added. We edited the caption to make this clearer. The ring is observed to slowly contract even in the control case due to a different simulation setup compared to the stationary ring in Figure 2 and 3. There are two main changes: (1) we increased the total actin concentration to 80 µM, and (2) we initialized a ring-like actin network rather than starting from a disordered network. Stationary rings are formed at a lower actin concentration (40 µM) and low myosin concentrations, but introducing additional myosin to this ring network leads to the formation of contractile clusters instead of a contracting ring. A higher actin concentration (80 µm) is required to obtain a contracting ring upon the introduction of additional myosin. To reduce computational costs, we simulated a pre-formed ring. However, the pre-formed ring at 80 µM actin exhibits a slow contraction. Although we cannot create a stationary ring under these conditions, we note that the control ring contracts significantly slower than upon the addition of excess myosin. We therefore believe that our results are in qualitative agreement with the experiments, given the complexity of the system. The details and rationale are discussed in Method section “Simulation setups of Latrunculin A, Calyculin A, and Y-27632 modeling”(pg. 34-35). It should be noted the actin rings in T cells also tend to slowly contract even under control conditions (some quantifications can be found in Figure 5c and Figure 5-supp.2a).

Also, in figure 1, the ring seems much more realistic than in the following.I am assuming the authors focused on a simpler system with fewer ingredients to establish the phase diagram, but this needs to be made clearer. Also, why not do a phase diagram with a simpler system, and a more realistic system to compare to experiments?For now, the comparison between experimental results and simulations is not as strong as claimed. Either the conclusions could be toned down a bit, or the discrepancies should be clearly discussed.

We agree that in vivo cellular systems are much more complex than we can simulate. There are two main reasons to focus on a simpler system: (1) our main goal is to explore the minimal requirements for actin ring formation, and (2) simulating more complex systems would have significantly higher computational costs. This high computational cost is also why we cannot work on a more realistic system (i.e. add Arp2/3 and increase actin concentration) to more directly compare with experiment. We have revised the section on the comparison between the simulation and experiment to highlight the limitations of the work, and proposed potential future improvements to the simulation in the Discussion section (pg. 19-20).

– MEDYAN licenseThis is not directly relevant to this article but falls under the public review guideline "the utility of the methods and data to the community".While MEDYAN being an open source software is a great boon for the community, it is crippled by its own license. The item 3 "Users can modify the MEDYAN source code for their own academic and research purposes, but cannot redistribute modified MEDYAN source code that differs in any way from Papoian lab's current MEDYAN distribution" goes against the core idea of open source scientific software: if a team decide to build upon MEDYAN, they will not be able to publish the modified code, and thus will not be able to share their results in a significant manner.Moreover, the guideline "cannot redistribute any other codes that use or extend any MEDYAN source code" prevents anyone from sharing wrappers and utilities for MEDYAN. Most scientific software uses the GPL (most restrictive) or MIT (most permissive) license with great success.Our main value as scientists is the production and sharing of knowledge – such a restrictive license does not seem to achieve this goal.

We thank the reviewer for bringing up this concern. First, we would like to note that highly influential scientific software come with licenses at all levels of permissiveness. For example, among broadly used molecular dynamics codes, Gromacs is issued under LGPL, OpenMM under MIT License and LGPL, while NAMD, AMBER and CHARMM have restrictive bespoke licenses (in case of AMBER and CHARMM, they are not even free for academic users). Nevertheless, all these software thrive and are productively used, where the nature of the license may influence some users’ decision on which molecular dynamics code to use.

In the context of mesoscale biomolecular simulations, Cytosim and Affines are GPL, while MEDYAN has a more restrictive license, giving choices to the users who are concerned about the license issue. Interestingly, for over 10 years (2007-2016) Cytosim was closed sourced, with only the executable being available, until it was open sourced in 2016.

To address the substance of the reviewer’s concern: we certainly would love for the users to write and distribute “wrappers and utilities for MEDYAN”. We will explore how MEDYAN’s license can be changed to encourage these possibilities and ask the tech transfer office to help. We would like to point out that the intellectual property for MEDYAN does not belong to our laboratory. It belongs to the University of Maryland, which has a tech transfer office that makes final decisions on software licenses based on an internal review involving attorneys, taking into account, in particular, the commercial potential for the software, among other factors.

We are hoping that the above discussion shows that choosing a license for a complex scientific software, such as MEDYAN, is a difficult endeavor, involving various legal and technical concerns. With that said, we will do our best to increase permissiveness of MEDYAN’s license to address some of the reviewer’s concerns.

– Patches as a metastable state: this is a bit confusing or over-stated. First, this is a highly out-of-equilibrium active system, and I do not think "metastable" quite applies. Both are fundamentally unstable states. A more likely claim is that treadmilling allows for a faster relaxation of filament elastic energy.

To avoid confusion, we have removed the word metastable from the introduction. However, we would like to note that in the context of Prigogine’s framework of dissipative structures, some structures may represent long-lived states that eventually transition to a final steady state structure. In this sense, we think of these long-live cluster states as metastable. In the revised text, we mentioned this point in the Discussion section (pg. 19).

B] Additional comments:– This is a very well-written article that is highly interesting – yet easy to read.– The claim that ring organization is not understood, not found in vitro, is far-fetched. First, actin rings in vitro have been observed, e.g. Miyazaki et al. 2015 to name one. Second, simulation of actin rings exists, cf Vavylonys 2008, Hang 2015, Nguyen 2018, Koudehi 2016; see also Dmitrieff 2017 for a ring of microtubules. In all these systems, confinement and filament length plays a major role in a ring formation, but the interplay between the rate of treadmilling and motor activity has not yet been really discussed to my knowledge. This is why not only the mean, but also the distribution of filament length has to be documented, and the role of confinement has to be discussed. The existing literature has to be more discussed.

We agree with the reviewer: the assembly of actin rings has been observed in vitro and *in silico*, but these rings typically require filaments (or filament bundles) longer than the confinement dimension or require membrane filament tethering. We have added these references and revised the manuscript to discuss these points (pg. 20).

– The difference in filament orientation in Figure 2 supp 4 is striking. I suspect that this could make for a much more striking phase diagram (by computing some order parameter for instance) than the current phase diagram.

This is an interesting point, but we believe that filament orientation is less important in the cluster case. Filament orientation is vital when located at the network periphery, where filaments perpendicular to the boundary will quickly depolymerize due to boundary exclusion and the Brownian Ratchet effect. However, many clusters are located far away from the boundary such that filaments can maintain treadmilling regardless of their orientation. That is why we believe a phase diagram of filament orientation may not be particularly informative.

Reference:

1. Svitkina, T. M. and Borisy, G. G. Arp2/3 complex and actin depolymerizing factor/cofilin in dendritic organization and treadmilling of actin filament array in lamellipodia. *J. Cell Biol.* 145, 1009–1026 (1999).

2. Takenawa, T. and Suetsugu, S. The WASP-WAVE protein network: connecting the membrane to the cytoskeleton. *Nat. Rev. Mol. Cell Biol.* 8, 37–48 (2007).

3. Kiuchi, T., Nagai, T., Ohashi, K. and Mizuno, K. Measurements of spatiotemporal changes in G-actin concentration reveal its effect on stimulus-induced actin assembly and lamellipodium extension. *J. Cell Biol.* 193, 365–80 (2011).

4. Wu, J.-Q. and Pollard, T. D. Counting Cytokinesis Proteins Globally and Locally in Fission Yeast. *Science (80-. ).* 310, 310–314 (2005).

5. Dominguez, R. and Holmes, K. C. Actin structure and function. *Annu. Rev. Biophys.* 40, 169–86 (2011).